# Pixel-Level Residual Diffusion Transformer: Scalable 3D CT Volume Generation

**Zhenkai Zhang, Markus Hiller, Krista A. Ehinger, Tom Drummond**
School of Computing and Information Systems, The University of Melbourne
zhenkaiz@student.unimelb.edu.au
{m.hiller, kris.ehinger, tom.drummond}@unimelb.edu.au

## Abstract

Generating high-resolution 3D CT volumes with fine details remains challenging due to substantial computational demands and optimization difficulties inherent to existing generative models. In this paper, we propose the Pixel-Level Residual Diffusion Transformer (PRDiT), a scalable generative framework that synthesizes high-quality 3D medical volumes directly at voxel-level. PRDiT introduces a two-stage training architecture comprising 1) a local denoiser in the form of an MLP-based blind estimator operating on overlapping 3D patches to separate low-frequency structures efficiently, and 2) a global residual diffusion transformer employing memory-efficient attention to model and refine high-frequency residuals across entire volumes. This coarse-to-fine modeling strategy simplifies optimization, enhances training stability, and effectively preserves subtle structures without the limitations of an autoencoder bottleneck. Extensive experiments conducted on the LIDC-IDRI and RAD-ChestCT datasets demonstrate that PRDiT consistently outperforms state-of-the-art models, such as HA-GAN, 3D LDM and WDM-3D, achieving significantly lower 3D FID, MMD and Wasserstein distance scores[1].

## 1 Introduction

Synthesizing high-resolution 3D medical images, such as CT volumes, is crucial for supporting various clinical applications, including diagnosis, segmentation, and anomaly detection. However, existing generative models struggle to balance the fidelity of local details with global structural coherence, and often suffer from computational inefficiency or insufficient representation of high-frequency features. Such prior works are mainly categorized into GAN-based and Diffusion-based methods. GAN-based approaches, such as HA-GAN (Sun et al., 2022), MM-GAN (Sun et al., 2020) and 3D-StyleGAN (Hong et al., 2021), can effectively generate realistic local details. However, they often suffer from mode collapse and training instability, and their substantial memory requirements pose a significant challenge, particularly when processing high-resolution 3D volumes.

As a compelling alternative, diffusion-based models have recently demonstrated more stable training dynamics and higher image fidelity. Representative diffusion-based methods include the 3D-DDPM (Dorjsembe et al., 2022), latent diffusion models (LDM) (Khader et al., 2023; Pinaya et al., 2022; Rombach et al., 2022), wavelet diffusion models (WDM-3D) (Friedrich et al., 2024) and triplane diffusion models (Zhang et al., 2025). Nevertheless, current diffusion models often rely on convolutional architectures like U-Net (Ronneberger et al., 2015), which inherently limit their capability to capture global context and long-range dependencies that are crucial for accurately synthesizing coherent structures.

In the context of 3D medical imaging, these architectural limitations become even more pronounced. Because the volume of a voxel feature map grows cubically with resolution, deploying a deep U-Net directly on high-resolution 3D volumes not only incurs extremely high memory and compute costs, but also typically requires compromise strategies such as patch-based processing, downsampling, or latent-space compression using VAE (Kingma & Welling, 2014) or VQ-VAE (Van Den Oord et al., 2017) methods. However, patch-based methods and downsampling inherently restrict the effective receptive field, compromising the model's ability to capture global anatomical coherence and long-range dependencies critical for clinical utility. In addition, methods relying on latent-space

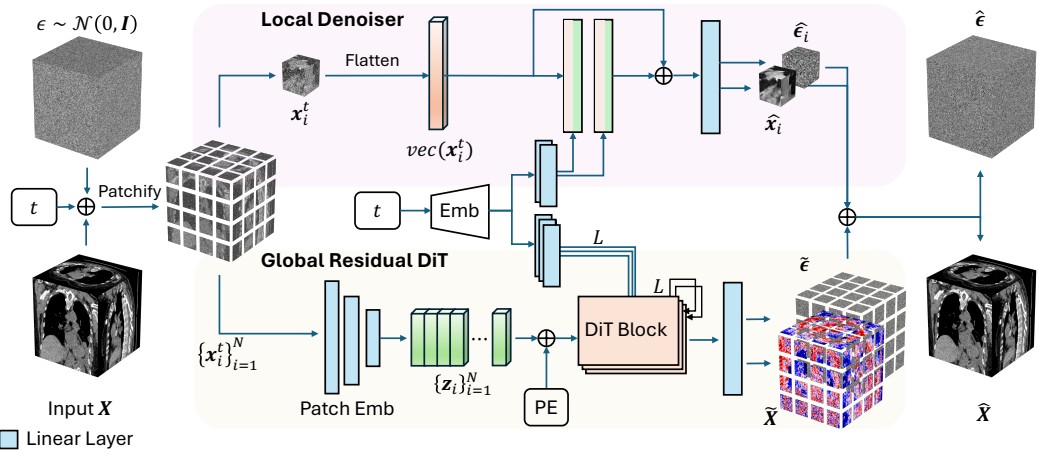

Figure 1: Overview of our *Pixel-Level Residual Diffusion Transformer* (PRDiT) model, composed of a **Local Denoiser** and a **Global Residual DiT**. The given input volume $\mathbf{X}$ is first divided into $N$ 3D patches. Each patch is flattened and passed through the 'local' multilayer perceptron (MLP) denoiser composed of Adaptive SwiGLU and Linear layers to provide *local* predictions $\hat{\mathbf{x}}_i$ and $\hat{\boldsymbol{\epsilon}}_i$. Complementing these 'local' predictions, the $L$ layer *Global* Residual Diffusion Transformer (DiT) module refines the overall prediction via residuals based on the global context. 'PE' indicates the positional encodings, which are only seen by the global residual DiT module.

compression often struggle to learn robust and representative features in the context of 3D medical imaging. The limited availability of training samples makes it difficult to adequately optimize the latent encoders, leading to poor reconstruction quality and the loss of critical anatomical details.

To address these limitations, we propose the *Pixel-Level Residual Diffusion Transformer* (PRDiT), a scalable diffusion transformer framework specifically designed for high-resolution 3D medical volume synthesis. PRDiT directly operates at the voxel level, bypassing the need for an autoencoder bottleneck, thus preserving essential details without compromising computational efficiency. The core idea is a two-stage residual learning strategy: a lightweight MLP-based Local Denoiser estimates coarse structures from overlapping 3D patches, and a Global Residual Diffusion Transformer with memory-efficient attention refines the remaining high-frequency residuals across the entire volume. To overcome the challenge that naively scaling Transformers to higher resolutions would increase the token count by $8\times$ and the attention cost by roughly $64\times$ when doubling the resolution, we further introduce a scaling strategy that reuses the pretrained low-resolution backbone. Focusing on only learning the missing high-frequency refinement at the higher resolution enables our model to perform high-resolution synthesis with minimal additional computational overhead.

The main contributions of this paper can be summarized as follows:

- We propose a two-stage diffusion transformer framework that directly synthesizes high-resolution 3D medical volumes at voxel-level, eliminating the need for an autoencoder bottleneck and thereby effectively preserving critical anatomical details.

- We demonstrate that using '*hot*' diffusion sampling to introduce controlled stochasticity during generation helps to improve sample diversity and reconstruction fidelity by balancing deterministic guidance with adaptive noise injection.

- We show that PRDiT scales efficiently to higher resolutions by reusing pretrained low-resolution components, substantially reducing training cost compared to coventional training on higher resolution datasets.

- We demonstrate the scalability and effectiveness of our PRDiT model through extensive experiments on the LIDC-IDRI (Armato III et al., 2011) and Rad-ChestCT (Draelos et al., 2020) datasets, achieving substantial improvements over state-of-the-art baselines across multiple quantitative metrics including 3D FID, MMD, and Wasserstein distances.

Figure 2: Detailed structure of our *Local Denoiser* module. The time embedding is used to modulate the local predictions via adaptive layer normalization (Peebles & Xie, 2023).

## 2 RELATED WORK

Generative models have significantly advanced our ability to synthesize realistic medical images, helping to alleviate data-scarcity and class-imbalance issues and improving the performance of downstream tasks such as segmentation and diagnosis. Existing research has explored GAN-based methods for generating 3D medical images. Kwon et al. (2019) combine an auto-encoder with a GAN model to generate 3D brain MRI samples, while Hong et al. (2021) tackle this problem by extending StyleGAN2 (Karras et al., 2020). HA-GAN (Sun et al., 2022) instead employs a hierarchical patch-based generator and discriminator to produce high-resolution 3D medical images.

Despite the promising works, GAN-based models still suffer from a range of issues including mode collapse, substantial memory requirements and low generation quality. Consequently, recent research has increasingly focused on diffusion models as a promising alternative, due to their increased training stability and higher fidelity in generated samples. 3D-DDPM (Dorjsembe et al., 2022) introduce denoising diffusion probabilistic models tailored for 3D medical data to generate high-resolution brain MRI scans. However, building diffusion models directly on 3D medical data incurs significant computational and memory costs, limiting scalability to high-resolution volumes. Following this, several new models propose corresponding improvements. Khader et al. (2023) combine a VQ-GAN-based latent space with a denoising diffusion probabilistic model to achieve high-quality synthesis of multi-modal 3D medical images, effectively enhancing the clarity and diversity of the generated images. Wavelet Diffusion Model (WDM) (Friedrich et al., 2024) improves efficiency by applying diffusion to wavelet-decomposed images, enabling high-resolution synthesis with reduced memory demands. TCAM-Diff (Zhang et al., 2025) incorporates a triplane-aware cross-attention mechanism that efficiently generates high-resolution 3D medical images while significantly reducing memory usage. However, all these models are based on variants of the convolutional U-Net architecture (Ronneberger et al., 2015), and excel at capturing local features but are limited in modeling crucial long-range dependencies.

To address this limitation in 2D tasks, Diffusion Transformer (DiT) (Peebles & Xie, 2023) models have recently been proposed, replacing convolutional patch embedders with Transformer blocks to better capture global context and long-range dependencies in image generation. DiT has demonstrated state-of-the-art performance in 2D image synthesis and has also shown promise when adapted to sparse 3D settings such as voxelized point clouds (Mo et al., 2023). However, despite these advances, we found that DiT still encounters substantial challenges in dense 3D scenarios, including unstable training behavior and optimization difficulties, in addition to the high computational cost required for scaling. Motivated by this, our work introduces a Residual Diffusion Transformer that decomposes the generation process into two stages: a lightweight MLP-based local denoiser operating on 3D patches, followed by a Transformer module that models and refines the global residuals. This two-stage design effectively balances computational efficiency with enhanced capability to capture both fine local details and global structural information, enabling scalable and high-fidelity synthesis of high-resolution 3D medical volumes.

## 3 SCALABLE 3D CT VOLUME GENERATION WITH PRDIT

When processing high-resolution 3D medical images, training a Transformer-based diffusion model to capture both low- and high-frequency components can quickly become prohibitively expensive and time-consuming. To overcome this challenge, we introduce our *Pixel-Level Residual Diffusion Transformer* (PRDiT) in this section. As illustrated in Figure 1, PRDiT decomposes the 3D diffusion process into two distinct branches: a *Local Denoiser* and a *Global Residual DiT* model. The local

denoiser estimates the 'local' content of each 3D patch independently to provide a prior estimate of both signal and noise of the 3D volume purely based on within-patch information, and is described in detail in Section 3.1. The residual diffusion Transformer's job is then to compute a refinement to these predictions by leveraging its global receptive field to correct any remaining errors across patch boundaries, as detailed in Section 3.2. We detail our *predictor-corrector* diffusion sampling method in Section 3.3, before introducing an efficient way to scale our 3D generative architecture to higher resolutions with only minimal computational overhead in Section 3.4. All code required to train and evaluate PRDiT will be made publicly available[1].

## 3.1 THE LOCAL DENOISER

Given an input volume $\boldsymbol{X} \in \mathbb{R}^{C \times H \times W \times D}$, we extract $N$ volumetric patches $\{\boldsymbol{x}_i \in \mathbb{R}^{C \times p \times p \times p}\}_{i=1}^N$ using a sliding window of size $p$ and stride $s < p$ to allow overlap between neighboring patches. This ensures that adjacent patches share contextual information at their borders, which we found to improve continuity in the reconstructed volume and helps to mitigate boundary artifacts (see Table 4 (a)). Each patch is then vectorized (i.e., flattened) to $\boldsymbol{v}_i = \text{vec}(\boldsymbol{x}_i) \in \mathbb{R}^d$, with $d = C \times p^3$.

Following the forward diffusion process proposed by Zhang et al. (2023), we sample a timestep $t \in \{1, \ldots, T\}$ and compute the noisy input volume at this timestep as

$$\boldsymbol{v}_i^t = \cos(\frac{t}{T}\frac{\pi}{2})\,\boldsymbol{v}_i \; + \; \sin(\frac{t}{T}\frac{\pi}{2})\,\boldsymbol{\epsilon}_i, \quad \text{with} \;\; \boldsymbol{\epsilon}_i \sim \mathcal{N}(0, I). \tag{1}$$

At each diffusion timestep $t$, we first compute a base embedding $\boldsymbol{c} = \text{TimeEmbed}(t) \in \mathbb{R}^h$ which is shared between both branches to facilitate consistent temporal conditioning. We project $\boldsymbol{c}$ into a *branch-specific* time embedding for the local branch as $\boldsymbol{c}_{\text{local}} \in \mathbb{R}^d$ using a linear layer, and pass it to the AdaLN-modulation network (Peebles & Xie, 2023) to produce the LayerNorm's shift/scale parameters $(\boldsymbol{\gamma}_1, \boldsymbol{\beta}_1, \boldsymbol{\gamma}_2, \boldsymbol{\beta}_2)$. As illustrated in Figure 2, these parameters allow our denoiser to adapt its processing of the volumetric patches based on the provided timestep for both refinement steps

$$\begin{aligned} \boldsymbol{z}_1 &= \text{SwiGLU}\big(\boldsymbol{\gamma}_1 \odot \text{LayerNorm}(\boldsymbol{v}_i^t) + \boldsymbol{\beta}_1\big) \\ \boldsymbol{z}_2 &= \text{SwiGLU}\big(\boldsymbol{\gamma}_2 \odot \text{LayerNorm}(\boldsymbol{z}_1) + \boldsymbol{\beta}_2\big), \end{aligned} \tag{2}$$

which yields, after a residual skip connection $\boldsymbol{z} = \boldsymbol{z}_2 + \boldsymbol{v}_i^t$ followed by a linear projection and reshaping, the predictions of the denoised input patch $\hat{\boldsymbol{x}}_i \in \mathbb{R}^{C \times s^3}$ as well as the noise $\hat{\boldsymbol{\epsilon}}_i \in \mathbb{R}^{C \times s^3}$. Note that the reshaped outputs do no longer overlap but provide estimates at the exact same dimensions as the input image and noise – aligning our Local Denoiser's output predictions with the standard diffusion objective of jointly estimating clean signal and noise.

We train the denoiser by minimizing the per-patch loss

$$\mathcal{L}_{\text{local}} = \mathbb{E}_i \Big[ \|\hat{\boldsymbol{x}}_i - \boldsymbol{x}_i\|_2^2 \; + \; \|\hat{\boldsymbol{\epsilon}}_i - \boldsymbol{\epsilon}_i\|_2^2 \Big] \tag{3}$$

to ensure we efficiently capture the local structure of each patch and accurately estimate its noise. At inference time, the denoised patch predictions $\{\hat{\boldsymbol{x}}_i\}^N$ together with the corresponding noise estimates $\{\hat{\boldsymbol{\epsilon}}_i\}^N$ are reassembled to provide a prior estimate of the entire input volume's signal and noise. This allows our Global Residual Diffusion Transformer, which will be described in the following, to focus exclusively on correcting the outputs via 'global' residuals $\{\boldsymbol{x}_i - \hat{\boldsymbol{x}}_i\}^N$, thereby reducing overall learning complexity and training time.

## 3.2 THE GLOBAL RESIDUAL DIFFUSION TRANSFORMER

Complementing the Local Denoiser, which processes each volume-patch independently without any global information, our Global Residual Diffusion Transformer component leverages the '*global*' full-volume context by jointly attending to all patch embeddings through multi-head self-attention. Following the original 2D DiT (Peebles & Xie, 2023), our model inherits the scalable nature of Diffusion Transformers, meaning it can enhance generation quality at increased computational resources by adjusting the Transformer's depth, width, or patch size, without necessitating network redesign. This adaptability supports diverse resolutions and volumes of medical data.

---

[1]All code for training and evaluating PRDiT is available at https://github.com/Fredy-Zhang/PRDiT

We freeze the Local Denoiser and train our global component to produce residuals that refine the local predictions (see Figure 1) using the same per-patch diffusion loss

$$\mathcal{L}_{\text{global}} = \mathbb{E}_i\big[\|\tilde{\boldsymbol{x}}_i - \boldsymbol{x}_i\|_2^2 + \|\tilde{\boldsymbol{\epsilon}}_i - \boldsymbol{\epsilon}_i\|_2^2\big], \tag{4}$$

now with $\tilde{\boldsymbol{x}}_i = \hat{\boldsymbol{x}}_i + \Delta\hat{\boldsymbol{x}}_i$ denoting the refined patch and $\tilde{\boldsymbol{\epsilon}}_i = \hat{\boldsymbol{\epsilon}}_i + \Delta\hat{\boldsymbol{\epsilon}}_i$ the refined noise.

### 3.3 IMPROVING SAMPLE QUALITY VIA PREDICTOR-CORRECTOR DIFFUSION SAMPLING

During inference, PRDiT generates 3D CT volumes using a modified reverse-diffusion sampling method derived from the deterministic ('cold') gradient-update scheme in (Zhang et al., 2023). Our approach reformulates their gradient-based generative path into distinct *cold predictor* and *hot corrector* steps: predicting a $k$-step jump forward before correcting by a $k-1$ backwards step. We found this to significantly improve both stability and generative quality (see ablations Section 4.4). Note that $k$ does not need to be an integer value, but is kept to $k = 2$ across most of our experiments.

As outlined in Algorithm 1, given a diffusion schedule with $T$ timesteps, the predictor step advances $k$ timesteps forward by applying the following $k$-scaled gradient update:

$$\boldsymbol{x}_{t-k} = \boldsymbol{x}_t - k \cdot \nabla\Big(\cos(\beta_t)\hat{\boldsymbol{x}}_0 + \sin(\beta_t)\hat{\boldsymbol{\epsilon}}\Big), \quad k \geq 1, \tag{5}$$

with $\beta_t = \frac{t}{T}\frac{\pi}{2}$, and $\hat{\boldsymbol{x}}_0$ and $\hat{\boldsymbol{\epsilon}}$ are the estimated clean volume and noise component predicted by the model. When $k = 1$, this procedure reduces to the standard single-step gradient update used in conventional *cold* diffusion sampling which advances one single timestep. For $k > 1$, the method becomes a *hot* diffusion process when combined with our corrector, introducing fresh noise per update to increase the level of stochastic exploration.

Concretely, following the $k$-step predictor stage, we perform a correction through a $k-1$ backwards diffusion step to reintroduce controlled noise and refine the sample as

$$\boldsymbol{x}_{t-1} = \Gamma_t^{(k)}\,\boldsymbol{x}_{t-k} + \sqrt{1 - \big(\Gamma_t^{(k)}\big)^2}\,\boldsymbol{\epsilon}', \quad \text{with } \Gamma_t^{(k)} := \frac{\cos(\beta_{t-1})}{\cos(\beta_{t-k})} \tag{6}$$

and where $\boldsymbol{\epsilon}' \sim \mathcal{N}(0, I)$ is newly injected random noise. This predictor–corrector scheme balances deterministic guidance (cold predictor) with adaptive stochasticity (hot corrector), enabling PRDiT to better preserve fine structures while maintaining global coherence in the generated volumes.

---

**Algorithm 1:** Predictor-Corrector Sampling

    **Input:**    Initial sample $x_T \in \mathbb{R}^{B \times C \times D \times H \times W}$;
                     Diffusion model $f_\theta$ with $f_\theta(x_t, t) := (\hat{\epsilon}_t, \hat{x}_0^t)$;
                     Predictor step multiplier $k \geq 1$.
    **Output:**  Sample sequence $\mathcal{X} = (x_t)_{t=0}^T$ and predictions $\hat{\mathcal{X}}_0 = (\hat{x}_0^t)_{t=1}^T$.

1  **Initialize:** $\mathcal{X} \leftarrow x_T$, angle map $\beta_t \leftarrow \frac{\pi}{2}\frac{t}{T}$.
2  **for** $t = T, T-1, \ldots, 1$ **do**
3     $x_t \leftarrow \text{last}(\mathcal{X})$
4     $(\hat{\epsilon}_t, \hat{x}_0^t) \leftarrow f_\theta(x_t, t)$                     ▷ `Model predictions at time` $t$
5     $g_t \leftarrow \sin(\beta_t)\,\hat{x}_0^t - \cos(\beta_t)\,\hat{\epsilon}_t$    ▷ `Gradient direction in cosine plane`
6     $k \leftarrow \min(k, t)$                   ▷ `Avoid` $t - k < 0$`, when` $t$ `is small`
7     $\Delta\beta \leftarrow \beta_t - \beta_{t-k}$
8     $x_{t-k} \leftarrow x_t - \Delta\beta \cdot g_t$       ▷ `Predictor:` $k$`-step jump to time` $(t-k)$
9     $\Gamma_t^{(k)} \leftarrow \cos(\beta_{t-1})/\cos(\beta_{t-k})$    ▷ `Scaling for variance preservation`
10    $\epsilon' \sim \mathcal{N}(0, I)$
11    $x_{t-1} \leftarrow \Gamma_t^{(k)}x_{t-k} + \sqrt{1 - \big(\Gamma_t^{(k)}\big)^2}\,\epsilon'$   ▷ `Corrector: variance preservation`
12    $\mathcal{X} \leftarrow \mathcal{X} \cup \{x_{t-1}\}$
13 **end**
14 **return** $(\hat{\mathcal{X}}_0, \mathcal{X})$

---

### 3.4 EFFICIENTLY GENERATING 3D VOLUMES AT HIGHER RESOLUTIONS

Training Diffusion Transformers from scratch at high resolutions like $256^3$ is impractical under typical GPU budgets: moving from $128^3$ to $256^3$ increases the voxel/token count by $8\times$, and the quadratic cost of self-attention raises memory/compute by roughly $64\times$. In practice this forces tiny batch sizes, triggers frequent out-of-memory failures, and destabilizes optimization, which makes the end-to-end high-resolution training inefficient and brittle (and often infeasible). To mitigate this, we construct a high-resolution generator that reuses our already-trained low-resolution model. As illustrated in Figure 3, we train only an additional high-resolution residual refinement module, which operates locally on individual volume patches and focuses on recovering the missing high-frequency details. Given a high-resolution noisy input volume $X^{\mathrm{HR}}$, we first downsample it to the lower resolution, apply the trained low-resolution model, and then upsample the prediction back to high resolution. This yields our initial 'rough' estimates for both signal and noise to be refined.

A key design choice is the use of *nearest downsampling*. Unlike smoother methods such as trilinear interpolation, nearest downsampling avoids averaging pixel values, which would otherwise attenuate high-frequency noise and reduce the signal's overall energy. Preserving this energy is essential to maintain the noise statistics expected by the pretrained low-resolution model, ensuring the downsampled input aligns with the model's training distribution and thereby avoiding mismatches during denoising. For the upsampling step, we apply trilinear interpolation to the predicted clean low-resolution image $\hat{X}_0^{\mathrm{LR}}$ to obtain $\hat{X}_0^{\mathrm{HR}}$, as this smoother interpolation better preserves structural continuity and reduces artifacts in the anatomical features, yielding a more natural high-resolution initialization. Since the noise is inherently stochastic and discontinuous, we apply nearest upsampling to the predicted noise $\hat{\epsilon}^{\mathrm{LR}}$ to obtain $\hat{\epsilon}^{\mathrm{HR}}$, ensuring its energy level is preserved.

To recover details lost during down- and upsampling, we introduce a high-resolution residual refinement module based our previously-introduced Local Denoiser design (Figure 2). This module refines the upsampled signal and noise estimates from the low-resolution component using the set of noisy input volume-patches and information about the current time step as shown in Figure 3. We keep the already-trained low-resolution model frozen and train only the refinement module at high resolution, which helps it focus its capacity on fine structures rather than relearning global anatomy. Our training uses the standard signal-and-noise estimation objective on high-resolution patches (Equation (3)), augmented with a low-frequency consistency term that encourages the downsampled high-resolution predictions to match the low-resolution outputs. We found this simple constraint to help keeping large-scale appearance stable while allowing the residual module to allocate its capacity to high-frequency detail.

Unlike cascaded super-resolution methods that typically modify only the final sample and often disrupt the low-frequency structure learned during diffusion, our approach integrates the high-resolution residual refinement module directly within the sampling loop. At each step, we downsample the current state to query the low-resolution model, upsample its estimate, and immediately refine high-frequency details before proceeding. This in-loop strategy yields sharp textures, reduces edge artifacts and promotes more stable detail formation throughout the diffusion process.

## 4 EXPERIMENTS

We start with a brief overview of the datasets, related baselines and metrics that are used for evaluation. We then present a number of qualitative and quantitative insights to demonstrate our approach's advantages in unconditional volume synthesis over related methods. Implementation details are provided in Appendix A.9, and additional experimental results can be found in Appendix A.11.

**Datasets.** We evaluate the performance of our model on two publicly available 3D medical imaging datasets: LIDC-IDRI (Armato III et al., 2011) and Rad-ChestCT (Draelos et al., 2020). LIDC-IDRI contains 1,018 thoracic CT scans with annotations of lung nodules from multiple expert radiologists. The scans exhibit varied resolutions and patient anatomies. For preprocessing, we clip the intensity values to the lung window range, resample all volumes to an isotropic voxel spacing of 1 mm, and crop or pad them to a uniform size of $256 \times 256 \times 256$. For experiments involving lower resolutions, volumes are downsampled via average pooling, then normalized to $[-1, 1]$. Rad-ChestCT includes 3,630 chest CT scans from 1,800 adult patients, representing approximately $10\%$ of the full dataset, which contains 35,747 scans from 19,661 patients. The dataset encompasses a broad spectrum

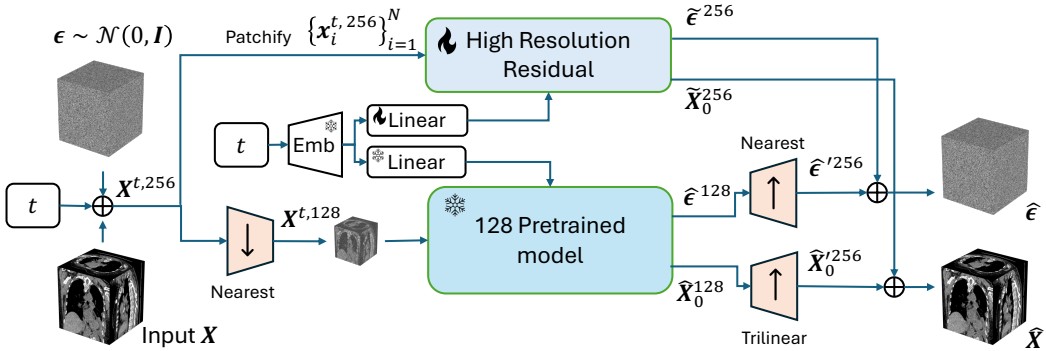

Figure 3: Our high-resolution PRDiT$\uparrow^{256}$ model integrates the pretrained and frozen lower-resolution PRDiT$\uparrow^{128}$ core to serve as a structural prior. Scaling the $128^3$ base resolution through trilinear upsampling allows its dedicated High-Resolution Residual path to capture fine-grained details in a parameter-efficient manner to generate high-quality $256^3$ outputs.

of thoracic diseases and anatomical variations. We center-crop to a fixed field of view, resize to $256 \times 256 \times 256$, optionally downsample, and normalize to $[-1, 1]$ for consistency with LIDC-IDRI.

**Baseline Models.** We contrast our PRDiT to three recently published 3D medical image generation models. WDM-3D (Friedrich et al., 2024), a diffusion model that decomposes each 3D volume into multi-resolution subbands via discrete wavelet transforms, denoises each subband with a 3D U-Net, and then reconstructs the full volume with an inverse wavelet transform. 3D LDM (Khader et al., 2023), a latent diffusion model that employs a VQ-GAN encoder to map volumes into a compact latent space, performs diffusion in that latent space and reconstructs volumes with a matching 3D convolutional decoder. HA-GAN (Sun et al., 2022) in contrast uses a hierarchical GAN whose generator comprises a low-resolution "global" and a high-resolution "local" 3D convolutional network, each guided by multi-scale discriminators on corresponding sub-volumes.

**Evaluation Metrics.** We evaluate unconditional generative quality using the 3D Fréchet Inception Distance (FID), Maximum Mean Discrepancy (MMD) (Gretton et al., 2012; Sun et al., 2022) and the pairwise Wasserstein distance using WGAN (Arjovsky et al., 2017; Zhang et al., 2025). Following common practice, we extract volumetric embeddings from both real and generated CT volumes using a pretrained Med3D encoder (Chen et al., 2019), then quantify their distributional discrepancy by fitting multivariate Gaussians and computing the 3D FID as well as by applying an RBF-kernel MMD (Gretton et al., 2012) in the same feature space. Lower 3D FID and MMD values indicate that the generated feature distribution more closely matches the real distribution. The Wasserstein distance is obtained by comparing the WGAN critic's scores on generated samples against those on real data, following the evaluation approach of Zhang et al. (2025). A lower Wasserstein distance indicates that the critic assigns similar scores to generated and real volumes, implying the generated distribution closely matches the true data distribution and thus higher sample fidelity. We perform pairwise comparisons to one 'anchor' model. See Appendix A.9.3 for additional details.

### 4.1 CONTRASTING PRDiT TO THE STATE-OF-THE-ART

To demonstrate the performance of our PRDiT, we compare three model variants (4 layers, 8 layers and 12 layers) against the baselines HA-GAN (Sun et al., 2022), 3D-LDM (Khader et al., 2023) and WDM-3D (Friedrich et al., 2024) on both LIDC-IDRI (Armato III et al., 2011) and RAD-ChestCT (Draelos et al., 2020) and report the results in Table 1. Our smallest 4 layer PRDiT-4L variant already achieves lower 3D FID and MMD scores than its competitors across both datasets, outperforming them across all metrics despite its shallow depth. Increasing PRDiT's depth to 8 and 12 layers yields consistent improvements in generative quality and demonstrates the scalability of our approach; albeit at increased computational cost. Note, however, that all PRDiT variants reuse the same Local Denoiser module which only has to be trained once, effectively reducing the computational overhead and training time when scaling our architecture to greater depths.

Following Zhang et al. (2023), we additionally report the pairwise W-Scores with our PRDiT-12L as the 'reference' model. This score effectively quantifies the ratio between the distances of each

model's distribution of generated samples to that of the real data (see Appendix A.9.3). A score of '1.0' indicates equal generative quality between two models, while values greater than 1 reflect degraded performance relative to the reference - reaffirming the quality of even our 4-layer model.

## 4.2 COMPARING PERCEIVED IMAGE QUALITY – SAGITTAL SLICES

Figure 5 shows sagittal slices from each generative method alongside real CT scans, providing a qualitative comparison between our PRDiT model and its competitors. HA-GAN (Sun et al., 2022) notably blurs bone contours and introduces unnatural texture artifacts throughout the volume. 3D-LDM (Khader et al., 2023) in contrast demonstrates clearer structural details and reduced noise, however still suffers from insufficient anatomical detail and streak-like artifacts. WDM-3D (Friedrich et al., 2024) further improves clarity and reduces overall artifacts, but continues to exhibit noticeable blocky noise and unclear bone boundaries. Our residual DiT model substantially addresses these shortcomings by delivering sharper bone edges, smoother and more consistent organ

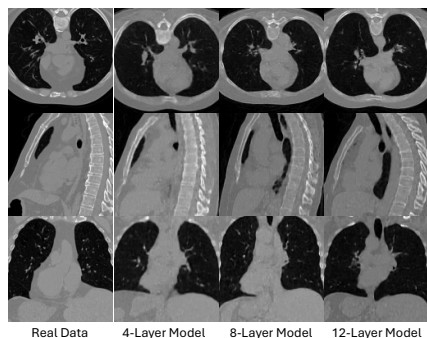

Real Data    4-Layer Model    8-Layer Model    12-Layer Model

Figure 4: Variation of image quality across different depths of PRDiT.

boundaries, and clearer anatomical details. These improvements highlight our method's capability to more accurately reconstruct subtle features present in real medical scans. In addition, increasing model depth yields progressively sharper cortical bone edges, clearer bronchial walls and vessel bifurcations, and smoother soft–tissue boundaries (Figure 4). These qualitative gains align with the results in Table 1, which show a consistent decrease in FID and MMD as depth increases.

## 4.3 EFFICIENTLY SCALING UP TO HIGHER RESOLUTIONS

We contrast our efficient high-resolution PRDiT against its competitors using the LIDC-IDRI (Armato III et al., 2011) dataset at $256^3$ resolution. As the results in Table 2 show, our PRDiT-4L$\uparrow^{256}$ clearly outperforms the other methods on both FID and MMD while incurring substantially lower training costs. While training 3D-LDM proved infeasible in our setup due to memory constraints, WDM-3D exhibited pronounced sensitivity to random seeds.

Table 2: Results on LIDC-IDRI at $256^3$ resolution. Reported GPU-hours are measured on an A100. FID values are scaled by $10^3$ (as is common practice).

| Model | FID↓ | MMD↓ | Training Cost |
|---|---|---|---|
| HA-GAN | 3.98 | 0.2237 | 140 GPUh |
| 3D-LDM | OOM | OOM | - |
| WDM-3D | 5.60 | 0.2590 | 120 GPUh |
| PRDiT-4L$\uparrow^{256}$ | 2.28 | 0.1370 | 36 GPUh |

While most seeds yield high-quality samples, a small fraction of severely degraded outputs disproportionately inflates FID/MMD, resulting in high variance and reduced evaluation robustness. Generated images of all methods are provided in Appendix A.11.3 for visual inspection.

Table 1: Comparison of unconditional 3D generative methods on LIDC-IDRI (Armato III et al., 2011) and RAD-ChestCT (Draelos et al., 2020), evaluated on volumes of size $128^3$. FID values are scaled by $10^3$. Reported are mean $\pm$ std over three seeds. Pairwise W-Score (ratio) is computed using a trained WGAN-GP critic against PRDiT-12L as reference. See Appendix A.9.3 for details.

| Model | LIDC-IDRI | | | RAD-ChestCT | | |
|---|---|---|---|---|---|---|
| | FID↓ | MMD↓ | W-Score↓ | FID↓ | MMD↓ | W-Score↓ |
| HA-GAN | $3.26 \pm 0.27$ | $0.2071 \pm 0.004$ | $5.42 \pm 1.46$ | $3.92 \pm 0.38$ | $0.183 \pm 0.015$ | $2.48 \pm 0.14$ |
| 3D-LDM | $7.62 \pm 0.21$ | $0.3458 \pm 0.006$ | $2.62 \pm 0.28$ | $4.14 \pm 0.74$ | $0.228 \pm 0.017$ | $2.87 \pm 0.36$ |
| WDM-3D | $3.67 \pm 0.38$ | $0.1885 \pm 0.017$ | $1.31 \pm 0.26$ | $4.11 \pm 0.58$ | $0.213 \pm 0.039$ | $1.27 \pm 0.11$ |
| PRDiT-4L (ours) | $2.04 \pm 0.18$ | $0.1852 \pm 0.009$ | $1.23 \pm 0.08$ | $1.92 \pm 0.28$ | $0.169 \pm 0.010$ | $1.21 \pm 0.04$ |
| PRDiT-8L (ours) | $1.86 \pm 0.12$ | $0.1650 \pm 0.011$ | $1.14 \pm 0.13$ | $1.62 \pm 0.07$ | $\mathbf{0.149} \pm 0.010$ | $1.02 \pm 0.02$ |
| PRDiT-12L (ours) | $\mathbf{1.41} \pm 0.17$ | $\mathbf{0.1501} \pm 0.010$ | $\mathbf{1.00}$ | $\mathbf{1.45} \pm 0.21$ | $0.159 \pm 0.007$ | $\mathbf{1.00}$ |

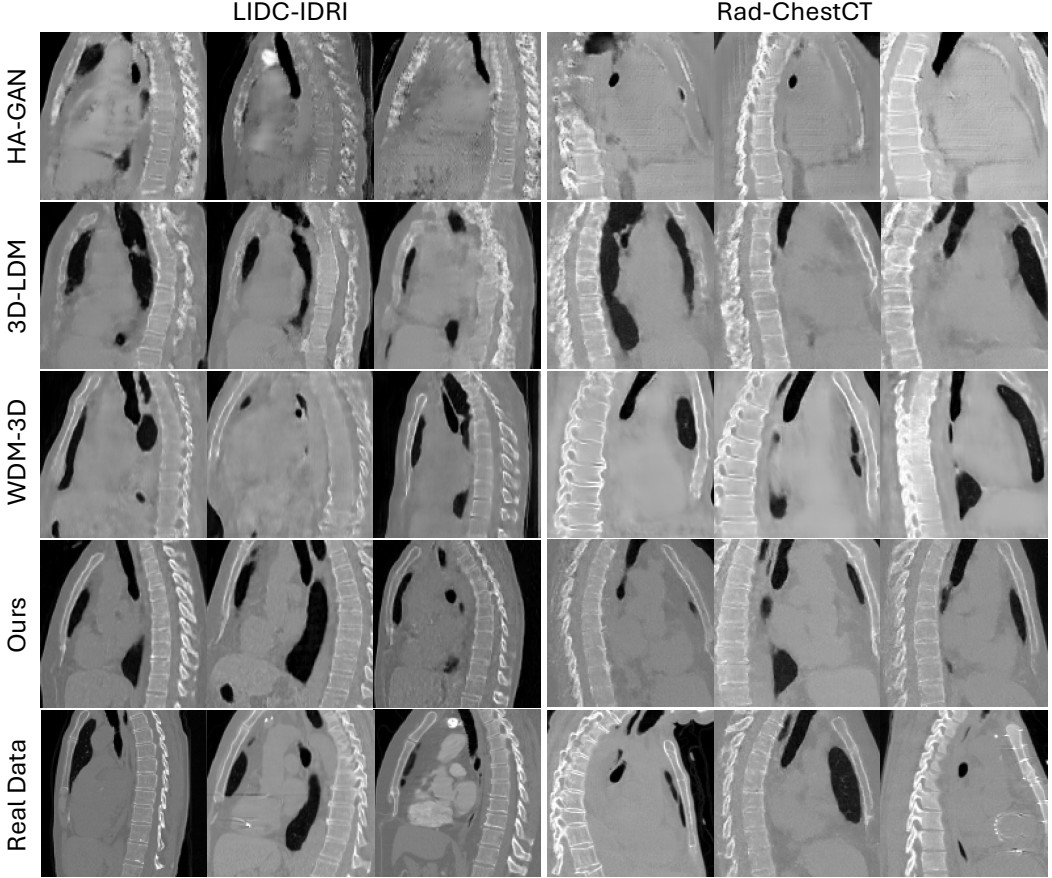

Figure 5: Qualitative comparison of unconditional 3D generative methods on the LIDC-IDRI (Armato III et al., 2011) and Rad-ChestCT (Draelos et al., 2020). From top to bottom, the rows correspond to (1) HA-GAN (Sun et al., 2022), (2) 3D-LDM (Khader et al., 2023), (3) WDM-3D (Friedrich et al., 2024), (4) our method, and (5) real medical data. Our approach better preserves fine anatomical details such as vertebral bone edges and organ boundaries while suppressing spurious artifacts compared to prior models. Additional examples are provided in Appendix A.11. *Note: Images are independently generated and unconditional, and cannot be compared column-wise. Regions appearing as 'black voids' within the body are low-density air cavities that are also present in the real data (see Appendix A.4, Figure 6 for more details).*

To efficiently validate the quality our high-resolution upsampling strategy against expensive from-scratch training, we additionally train a PRDiT model at $64^3$ and upgrade to $128^3$ using our proposed high-resolution upsampling methodology, and compare its results to the PRDIT model trained from scratch on $128^3$. While training from scratch does indeed lead to slightly improved performance across FID and MMD, our efficient variant is more than 6 times faster in training – providing a convincing tradeoff in terms of quality-to-compute (Table 3).

Table 3: Comparison of high-resolution training strategies on LIDC-IDRI. FID scaled by $10^3$.

| Method | FID↓ | MMD↓ | Training Cost |
|---|---|---|---|
| PRDiT$_{128}^{\text{scratch}}$ | 2.04 | 0.1853 | 80 GPUh |
| PRDiT$_{64}\uparrow^{128}$ | 2.89 | 0.1893 | 12 GPUh |

## 4.4 ABLATION EXPERIMENTS

**Contributions of Individual Design Choices.** We conduct ablation experiments on the LIDC-IDRI (Armato III et al., 2011) dataset using our PRDiT-4L model to assess the contributions of our design choices. Table 4 (a) shows that removing the overlap between patches significantly worsens 3D FID and MMD scores due to boundary inconsistencies. Training our model without

Table 4: Ablation studies using our 4-layer variant PRDiT-4L on the LIDC-IDRI dataset. The reported FID values are scaled by $10^3$, as is common practice. (a) Contribution of the architecture's individual components. (b) Effect of varying the predictor steps $k$ in our predictor-corrector setup.

<table>
<tr><td colspan="3" align="center">(a)</td><td colspan="3" align="center">(b)</td></tr>
<tr><td>Model variant</td><td>FID↓</td><td>MMD↓</td><td>$k$</td><td>FID↓</td><td>MMD↓</td></tr>
<tr><td>Full model</td><td>2.04</td><td>0.1853</td><td>1.0 (cold)</td><td>8.889</td><td>0.3490</td></tr>
<tr><td>w/o overlap</td><td>3.27</td><td>0.2304</td><td>2.0</td><td>2.173</td><td>0.1849</td></tr>
<tr><td>w/o local denoiser</td><td>3.10</td><td>0.2174</td><td>3.0</td><td>3.112</td><td>0.2112</td></tr>
<tr><td>w/o global DiT</td><td>41.92</td><td>0.7795</td><td>4.0</td><td>4.184</td><td>0.2425</td></tr>
</table>

the Local Denoiser (i.e., DiT-only) also markedly reduces the performance – whereas using *only* the local component without any global refinement leads to the worst results.

**Hot-vs-Cold Diffusion.** We evaluate the effect of varying the predictor step $k$ of our PRDiT-4L variant model using the LIDC-IDRI (Armato III et al., 2011) dataset. The results in Table 4 (b) demonstrate that the transition from 'cold' ($k = 1$) to 'hot' ($k > 1$) significantly enhances generative output performance, with $k = 2$ achieving the best results in this setup. Note that the number of steps is technically not limited to integers, and future work might explore whether further optimization along this axis might yield even greater quality improvements.

To clearly highlight the differences between our proposed predictor-corrector diffusion sampling method and conventional sampling techniques ('cold' and 'hot'), we provide detailed discussion as well as a range of additional insights and comparisons in Appendix A.6, including both quantitative and qualitative results across a range of settings.

**Inference times.** To further assess PRDiT's suitability for practical, time-sensitive applications, we compare its inference speed against three competing models. All inference times are measured with a batch size of 1 on an A100 GPU at resolution $128^3$, with detailed results presented in Appendix A.7. It is worth noting that, due to its GAN-based architecture and non-iterative, single-pass image generation, HA-GAN (Sun et al., 2022) is by far the fastest, with an average inference time of just $0.01$s. Among the diffusion-based approaches, all PRDiT variants run markedly faster than the competitors, with inference times ranging from $11.5$s (4L) to $21.9$s (12L), compared to $26.1$s for 3D-LDM (Khader et al., 2023) and $34.6$s for WDM-3D (Friedrich et al., 2024).

## 5 CONCLUSION

In this paper, we have presented *PRDiT* – a novel and scalable residual Diffusion Transformer model which efficiently solves the 3D CT Volume generation task via two dedicated branches and a two-stage training strategy, consisting of: a Local Denoiser and a Global Residual Transformer. By decoupling the task of modeling coarse local details and refining high-frequency global residuals, PRDiT effectively reduces computational overhead. Extensive evaluations on LIDC-IDRI and Rad-ChestCT datasets demonstrate that our method achieves superior generative performance compared to state-of-the-art methods such as HA-GAN, 3D LDM and WDM-3D, consistently producing lower 3D FID, MMD, and Wasserstein distances. Our experiments further validate the model's scalability, illustrating consistent performance improvements with increasing transformer depth. These results highlight PRDiT's potential as a powerful and efficient method for synthesizing high-resolution 3D medical volumes, with promising applicability to various clinical and diagnostic scenarios.

## 6 ACKNOWLEDGMENT

We would like to thank Sun et al. (2022), Khader et al. (2023) and Friedrich et al. (2024) for making their code publicly available, enabling us to perform direct comparisons to our method.

This research was supported by The University of Melbourne's Research Computing Services and the Petascale Campus Initiative. MH was supported by the Australian Research Council (ARC) through grant DP230102775.

## 7 REPRODUCIBILITY STATEMENT

We have made every effort to ensure that the results presented in this paper are reproducible. We explain all datasets, data preprocessing, baseline models, evaluation metrics, and training details used in our method in experimental section. To support reproducibility, we include a detailed overview of our architecture in Figures 1,2,3, which fully describe the design. In addition, the experimental setup, including training steps, model configurations, and hardware details, are all described in detail in Appendix A.9. The 3D medical CT datasets, such as LIDC-IDRI and Rad-ChestCT, are publicly available, ensuring consistent and reproducible evaluation results. All code required for training and evaluating our method will be publicly released on github, as detailed in footnote 1 on page 4.

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

# A  Appendix

## A.1  LLM Usage

In this paper, Large Language Models (LLMs) were utilized as a general-purpose assist tool to aid and polish the writing process. We used the LLM to refine language and improve clarity across the Introduction, Related Work and Conclusion sections.

## A.2  Hallucination Risks and Potential Misuse

In the context of this work, 'hallucination' would refer to anatomically implausible or extremely rare structures that lie far outside the training distribution.

From a technical angle, we monitor this risk partly via the W-score metric: if the model frequently produced unrealistic volumes, the WGAN critic would assign larger distances between real and generated samples, and the W-score would deteriorate. In practice, our W-scores remain close to those of real data, which suggests that severe hallucinations are uncommon in our experiments.

That said, we want to emphasize that we do *not* position PRDiT as a tool for direct clinical decision-making. Synthetic CT volumes should be used only for research, simulation, or carefully controlled data augmentation, and always under expert supervision. Any downstream clinical application must rely on appropriate validation on real patient data and the involvement of qualified radiologists.

## A.3  Limitations and Future Work

Our current implementation is constructed upon a simple linear mapping, including patch embeddings, which we treat as the base structure. While effective, this design may limit the model's ability to capture complex local spatial correlations. A natural extension would be to incorporate convolutional neural networks (CNNs) for feature extraction, which could enhance representational richness and further improve generative quality. We leave this direction as an important opportunity for future exploration.

PRDiT inherits the high training cost of Transformer-based models, especially in handling full-volume 3D attention. Future work might explore 3D-specific optimizations, such as linear or windowed attention, to reduce computational demands. We also plan to extend PRDiT to conditional generation tasks, enabling guidance from segmentation masks or sparse inputs for broader clinical applications.

It is important to point out that there generally exists no voxel-wise 'ground truth' for novel unconditionally-generated volumes. This means that downstream segmentation or detection studies would require dense, expert annotations for both real and synthetic CT volumes, which are expensive and therefore unfortunately beyond the scope and resources of this work.

The objective of this paper is to make high-fidelity unconditional 3D CT generation feasible at scale and to rigorously evaluate distributional fidelity. We therefore (i) do not claim that immediate clinical deployment of our method or similar methods of this kind are feasible, and (ii) want to point out that such downstream and clinical evaluations are outside the scope of the present paper, and we have to leave a systematic study of using our model for downstream segmentation/detection and assessing its clinical impact as future work.

## A.4  Characteristics of the Training Datasets

In this section, we explain why some of the generated CT outputs appear to contain black 'voids' in the parts of the images, like the lower-left region within Figure 5. Careful inspection of the original datasets shows that these dark 'void' regions actually correspond to naturally occurring low-density air cavities that are also present in the real data, rather than artifacts introduced by the hot-diffusion sampler or hallucination issues, as illustrated in Figure 6. In addition, following standard practice in prior works, our pre-processing clamps HU values to a fixed window, which further accentuates such

air regions as dark areas in both real and generated scans. For easier debugging and later verification, we additionally overlay the case IDs from the LIDC-IDRI dataset onto the images.

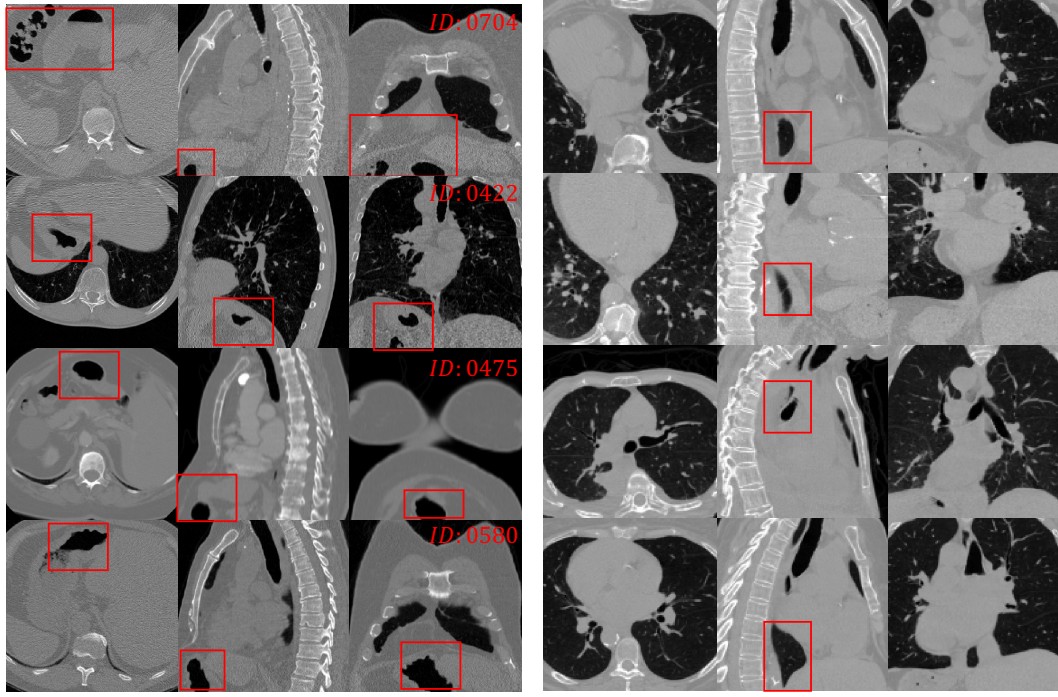

Figure 6: Ground truth CT samples from LIDC-IDRI (left) and Rad-ChestCT (right). Red boxes highlight naturally occurring low-density air cavities, showing that similar dark void regions are also present in the real data.

### A.5 THE LOCAL DENOISER

This section provides additional details for the local denoiser module. When processing high-resolution 3D medical images, training diffusion models with Transformers to capture both low- and high-frequency components is prohibitively expensive and time-consuming. As illustrated in Figure 1, we therefore decompose the learning process into two stages: a local denoiser estimates the content of each 3D patch locally (see Figure 2), and a diffusion model with Transformers focuses exclusively correcting these via global residuals.

Concretely, given an input volume

$$\boldsymbol{X} \in \mathbb{R}^{C \times H \times W \times D},$$

we extract $N$ cubic patches

$$\{\boldsymbol{x}_i \in \mathbb{R}^{C \times p \times p \times p}\}_{i=1}^{N}$$

using a sliding window of size $p$ and stride $s < p$ to allow overlap between neighboring patches, which ensures that adjacent patches share contextual information at their borders to improve continuity in the reconstructed volume and mitigate boundary artifacts. Each patch is vectorized as

$$\boldsymbol{v}_i = \text{vec}(\boldsymbol{x}_i) \in \mathbb{R}^d, \quad d = C \times p^3.$$

To follow the forward diffusion process (Zhang et al., 2023), we sample a timestep $t \in \{1, \dots, T\}$ and compute

$$\boldsymbol{v}_i^t = \cos(\frac{t}{T} \frac{\pi}{2}) \boldsymbol{v}_i \; + \; \sin(\frac{t}{T} \frac{\pi}{2}) \boldsymbol{\epsilon}_i, \quad \boldsymbol{\epsilon}_i \sim \mathcal{N}(0, I).$$

Given a diffusion timestep $t$, we first compute a shared base embedding

$$\boldsymbol{c} = \text{TimeEmbed}(t) \; \in \; \mathbb{R}^h.$$

We then project $c$ into a stage-specific time embedding for the local stage

$$c_{\text{local}} = W_c \, c + b_c \; \in \; \mathbb{R}^d,$$

and feed $c_{\text{local}}$ into the AdaLN modulation network to produce the LayerNorm shift/scale parameters

$$(\gamma_1, \beta_1, \gamma_2, \beta_2) = \text{AdaLN}(c_{\text{local}}) \; \in \; \mathbb{R}^{4d}.$$

In the global residual stage, we similarly compute

$$c_{\text{fine}} = W_f \, c + b_f \; \in \; \mathbb{R}^h,$$

and use $c_{\text{fine}}$ within each AdaLN layer of the DiT blocks, ensuring consistent temporal conditioning across both modules.

The signal is then refined via two SwiGLU-MLP blocks

$$z_1 = \text{SwiGLU}\big(\gamma_1 \odot \text{LayerNorm}(v_i^t) + \beta_1\big),$$
$$z_2 = \text{SwiGLU}\big(\gamma_2 \odot \text{LayerNorm}(z_1) + \beta_2\big),$$

and added residually as $z = z_2 + v_i$. A final linear projection produces

$$[\hat{v}_i, \; \hat{\epsilon}_i] = W_{\text{final}}(z) \; \in \; \mathbb{R}^{2d}, \quad \hat{v}_i, \; \hat{\epsilon}_i \in \mathbb{R}^d,$$

which is reshaped back to $\hat{x}_i \in \mathbb{R}^{C \times s^3}$. Here $\hat{v}_i$ is the denoised patch and $\hat{\epsilon}_i$ is the predicted noise, which aligns our MLP's outputs with the standard diffusion objective of jointly estimating clean signal and noise.

We train the denoiser by minimizing the per-patch loss

$$\mathcal{L} = \mathbb{E}_i\Big[\|\hat{x}_i - x_i\|_2^2 \; + \; \|\hat{\epsilon}_i - \epsilon_i\|_2^2\Big],$$

ensuring efficient capture of local patche structure and accurate noise estimation. At inference, the denoised and overlapping patches $\{\hat{x}_i\}^N$ are passed to the Residual Diffusion Transformer, which focuses exclusively on correcting the predictions via global residuals $\{x_i - \hat{x}_i\}$, thereby reducing overall learning complexity and training time.

## A.6 PREDICTOR-CORRECTOR SAMPLING

This section provides more detail on our sampling scheme introduced in Section 3.3, and shows how it differs from previous standard ancestral or 'DDIM' samplers.

### A.6.1 QUALITATIVE INSIGHTS FOR VARYING $k$

To provide further insight in addition to the quantitative analysis of varying $k$ in the main paper, we provide qualitative results across different $k$ values across axial, coronal, and sagittal views in Figures 7 8 9.

### A.6.2 RELATIONSHIP TO CONVENTIONAL 'HOT' DIFFUSION SAMPLING

In this section, we outline how our sampling procedure differs from standard ancestral or 'DDIM' samplers: it is tailored to the cosine-sine schedule (Zhang et al., 2023) and lets us inject a controlled amount of extra stochasticity per step, which we show empirically improves FID/MMD over both deterministic sampling ($k = 1$, Table 4 (b), main paper) as well as previous methods (following).

The conventional ancestral 'hot' sampler performs a single stochastic update from $x_t$ to $x_{t-1}$ using only the predicted noise $\epsilon$. Our sampler can be viewed as a predictor-corrector variant of this ancestral sampler: instead of one stochastic step, we take $k$ deterministic denoising sub–steps followed by $k - 1$ stochastic correction steps, which lets us inject a stronger but controllable amount of noise per outer step.

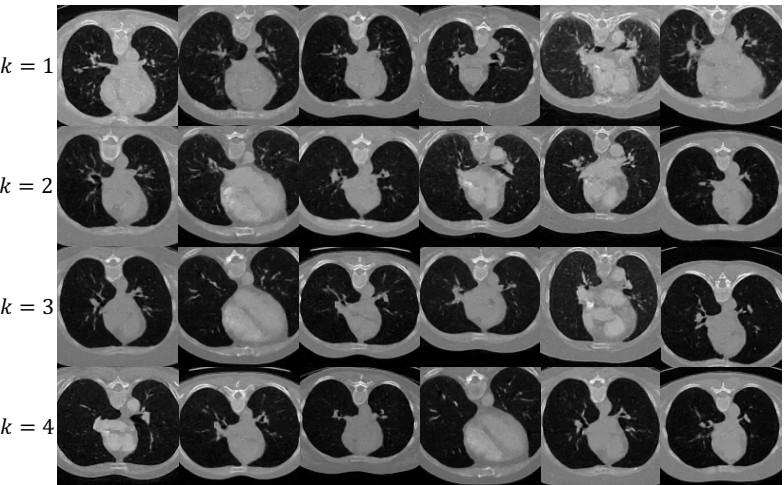

Figure 7: Qualitative comparison: axial view of generated chest CT samples for different $k$ values in LIDC-IDRI dataset.

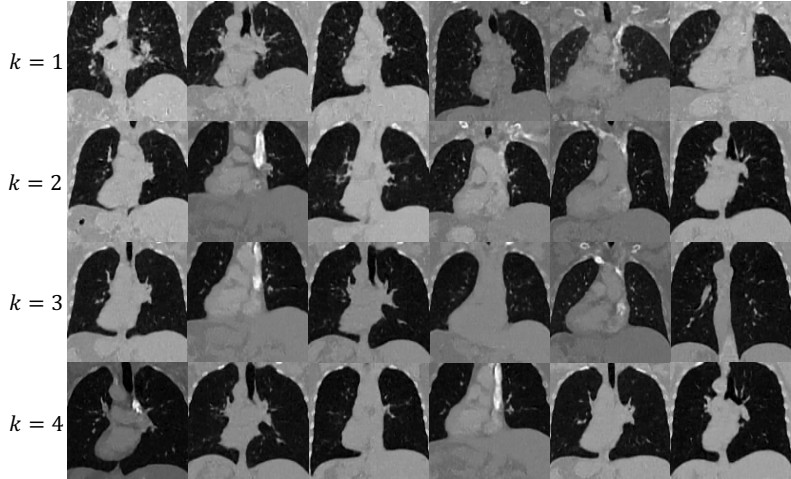

Figure 8: Qualitative comparison: coronal view of generated chest CT samples for different $k$ values in LIDC-IDRI dataset.

**Settings.** Our model in the main paper is trained to jointly predict the clean image $x_0$ and the noise $\epsilon$, whereas the conventional hot sampler only relies on $\epsilon$. For a fair comparison, we therefore refactor our generation process so that it also depends solely on the predicted noise. In this way, we get the $\hat{x}_0$ through

$$\hat{x}_0(x_t, t) = \frac{x_t - \sqrt{1 - \alpha_t}\epsilon_\theta(x_t, t)}{\sqrt{\alpha_t}},$$

which exactly follows the method from the conventional sampler. The conventional sampling method is defined as

$$x_{t-1} = \sqrt{\alpha_{t-1}}\,\hat{x}_0(x_t, t) + \sqrt{1 - \alpha_{t-1} - \sigma_t^2}\,\epsilon_\theta(x_t, t) + \sigma_t z \tag{7}$$

where $z \sim \mathcal{N}(0, I), \sigma_t = \sqrt{\frac{1 - \alpha_{t-1}}{1 - \alpha_t}}\sqrt{1 - \frac{\alpha_t}{\alpha_{t-1}}}, \sqrt{\alpha_t} = \cos(\beta_t), \sqrt{1 - \alpha_t} = \sin(\beta_t)$, which is the same sampling method used in DDPM or DDIM (setting $\eta = 1$).

Recalling Equations 5 and 6, our sampling method results in

$$\boldsymbol{x}_{t-1} = \Gamma_t^{(k)}\Big(\boldsymbol{x}_t - k \cdot \nabla\Big(\sqrt{\alpha_t}\hat{x}_0(x_t, t) + \sqrt{1 - \alpha_t}\epsilon_\theta(x_t, t)\Big)\Big) + \sqrt{1 - \big(\Gamma_t^{(k)}\big)^2}\, z, \tag{8}$$

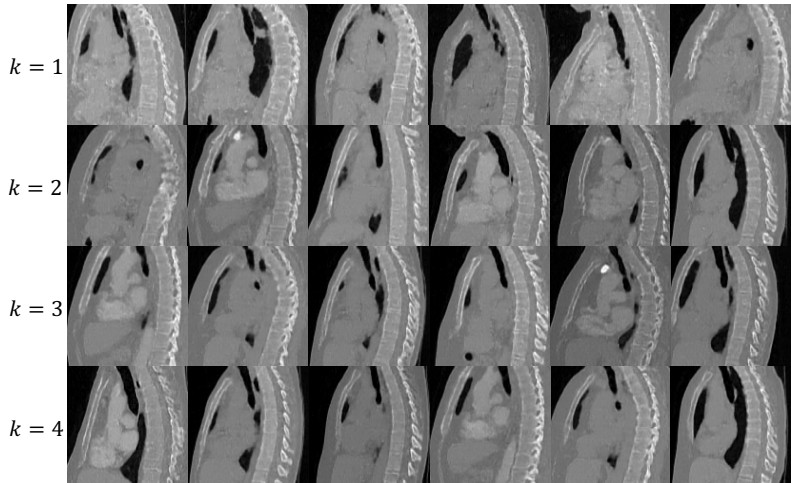

Figure 9: Qualitative comparison: sagittal view of generated chest CT samples for different $k$ values in LIDC-IDRI dataset.

| $k$ | FID↓ | MMD↓ |
|---|---|---|
| 1 (cold sampling) | 7.031 | 0.7285 |
| 2 | 6.132 | 0.6023 |
| 2.5 | 5.950 | 0.5893 |
| 3 | 5.736 | 0.5699 |
| 3.5 | **5.718** | **0.5682** |
| 4 | 6.960 | 0.6273 |
| Conventional (hot diffusion) | 6.456 | 0.6100 |

Table 5: Ablation experiments for *noise-only adaptation*: quantitative comparison on the number of predictor-corrector sub-steps $k$ in our *modified noise-only* sampler on LIDC-IDRI, compared to the conventional hot-diffusion sampler. FID is reported as $10^2$.

with $\Gamma_t^{(k)} = \frac{\sqrt{\alpha_{t-1}}}{\sqrt{\alpha_{t-k}}}$, where $k = \{1, 2, ..\}$, and $k$ is *not* restricted to integer values.

**Experiments.** To clearly highlight the differences between our proposed hot-diffusion sampling method and the conventional sampler, we provide both qualitative (Figure 10) and quantitative (Table 5) comparisons of their respective generation quality. The results show that our hot diffusion (e.g. $k = 3$) achieves better FID/MMD than both the deterministic case ($k = 1$) and the conventional hot-diffusion sampler, indicating that our proposed modified stochastic schedule is genuinely beneficial rather than just a reparameterization of the standard sampler.

## A.7  INFERENCE TIME

We quantify inference speed in Table 6 by measuring sampling time across three baseline models and our three variant models. All timings are reported as the mean of 10 runs with a batch size of 1.

## A.8  HIGH RESOLUTION MODELS ARCHITECTURE

The architectural structure of our high-resolution 3D CT image generation approach is provided in the body of the main paper in Figure 3. In this section, we additionally provide insights on the difference in image quality between our efficient high-resolution variant and expensive from-scratch training of PRDiT directly on higher-resolution images. Results are provided in Table 7 and demonstrate that while training from scratch does indeed lead to slightly improved performance across FID

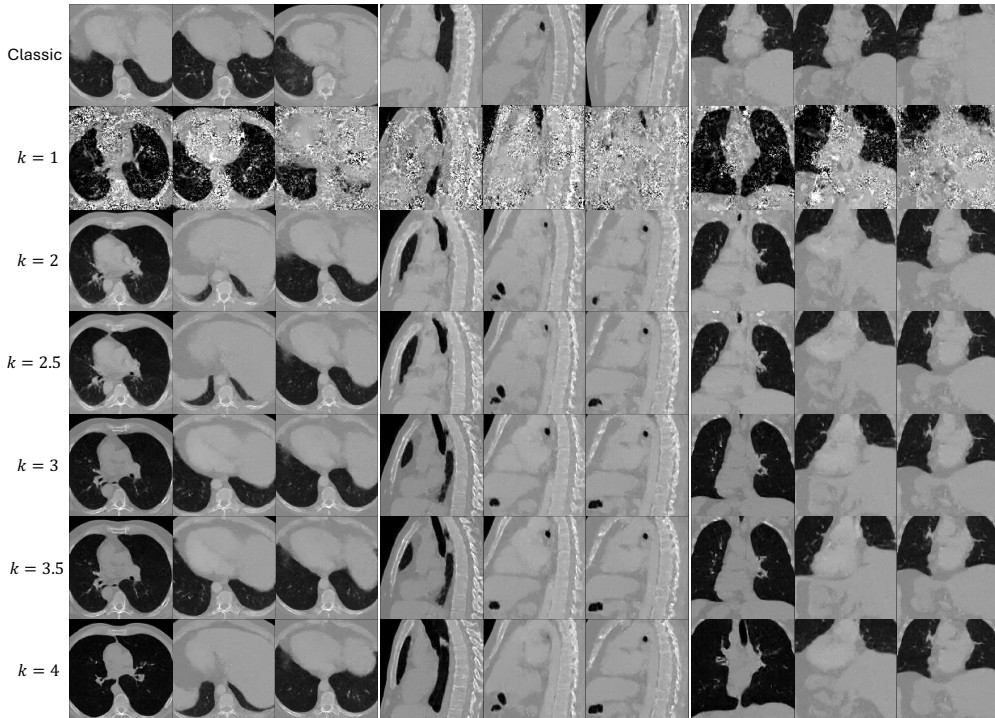

Figure 10: Ablation experiments: qualitative comparison on the number of predictor-corrector sub-steps $K$ in our *modified noise-only* sampler on LIDC-IDRI, compared to the conventional hot-diffusion sampler (classic in row 1).

Table 6: Inference sampling time per sample (seconds) with a batch size of 1 on an A100 GPU. Reported are mean $\pm$ std over 10 runs.

| Method | Mean (s) | Std (s) |
|---|---|---|
| HA-GAN (Sun et al., 2022) | 0.01020 | 0.00139 |
| 3D-LDM (Khader et al., 2023) | 26.0518 | 0.1009 |
| WDM-3D (Friedrich et al., 2024) | 34.6328 | 0.5915 |
| PRDiT-4L (ours) | 11.4590 | 1.1665 |
| PRDiT-8L (ours) | 18.3414 | 0.7329 |
| PRDiT-12L (ours) | 21.9311 | 0.8646 |

and MMD, our efficient variant is close to 7 times faster in training, providing a convincing tradeoff in terms of quality–to–compute.

Table 7: Ablation experiments: Comparison of high-resolution training strategies on LIDC-IDRI. We report a model trained directly at $128^3$ (Scratch) versus our progressive strategy that reuses a pretrained $64^3$ model and extends it to $128^3$ (Pretrained Low-to-High). FID is reported as $10^3$.

| Method | FID↓ | MMD↓ | Training Cost |
|---|---|---|---|
| PRDiT$_{128}^{\text{scratch}}$ | 2.04 | 0.1853 | 80 GPUh |
| PRDiT$_{64}\uparrow^{128}$ | 2.89 | 0.1893 | 12 GPUh |

In our high-resolution PRDiT$\uparrow^{256}$ model experiments, we freeze the entire low-resolution backbone, which includes both Local Denoiser and the Global Residual DiT modules, and only train a lightweight high-resolution refinement module. We made this choice on purpose: fine-tuning the full backbone at $256^3$ would be very memory- and compute-heavy, while the refinement module

can concentrate on adding the missing high-frequency details on top of a already well-trained low-resolution anatomy.

In practice, PRDiT-4L$\uparrow^{256}$ still outperforms all high-resolution baselines on LIDC-IDRI in both FID and MMD, and it does so with much lower GPU cost (Table 2). This suggests that freezing the backbone does not noticeably hurt performance at $256^3$. In addition, our comparison of from-scratch $128^3$ training with the progressive $64^3 \rightarrow 128^3$ variant (Table 3) shows only a small drop in FID/MMD for more than a $6\times$ reduction in training time, which further supports this design.

To provide further insight into how a frozen backbone compares against a jointly fine-tuned setup, we performed an ablation at $128^3 \rightarrow 256^3$, directly comparing a model with a frozen low-resolution backbone to one where we fine-tune the entire backbone. For the fine-tuned variant, we use a conservative learning rate of $1 \times 10^{-6}$ for 100 epochs to avoid disrupting the well-trained low-resolution weights, and evaluate it under exactly the same setup as the frozen-backbone model. As Table 8 shows, the fine-tuned backbone gives a modest but consistent improvement over the frozen one (better FID and MMD), indicating that full fine-tuning can help, but the gain is relatively small compared with the additional computational cost, which supports our choice to freeze the backbone in the main $256^3$ experiments.

To make these experiments reproducible, we detail below the exact configurations used to train all high–resolution models.

### A.8.1 TRAINING SETUP FOR HIGH–RESOLUTION MODELS

**Frozen–backbone settings.** We first freeze the entire low-resolution backbone (both the Local Denoiser and the Global DiT) and train only the high-resolution refinement module. The refinement operates on overlapping 3D patches of size $(12, 12, 12)$ with stride 8 and padding 2 voxels, matching the local denoiser configuration in the low-resolution model. We optimize with AdamW, using a learning rate of $6 \times 10^{-5}$, gradient clipping at 1.0, a batch size of 8 per GPU, and train for 2000 epochs.

**Full fine–tuning setting.** For the fine-tuning ablation, we start from the converged frozen-backbone high-resolution model and then unfreeze the entire pretrained backbone and high-resolution module, training all parameters jointly. To avoid disrupting the well-trained low-resolution weights, we use a much smaller learning rate of $1 \times 10^{-6}$, gradient clipping at 0.4, a batch size of 2 per GPU, and train for 100 epochs under otherwise the same setup.

### A.8.2 EXPERIMENTS RESULTS

As Table 8 and Figure 11 show, the fine-tuned backbone gives a modest but consistent improvement over the frozen one (better FID and MMD), indicating that full fine-tuning can help, but the gain is relatively small compared with the additional computational cost, which supports our choice to freeze the backbone in the main $256^3$ experiments.

Table 8: Ablation experiments: Comparison of high-resolution performance between frozen low-resolution model and fine-tune entire models. FID is reported as $10^3$.

| Model | FID$\downarrow$ | MMD$\downarrow$ |
|---|---|---|
| Frozen $128^3$ | 2.280 | 0.1370 |
| Fine-tuned $128^3$ | 1.702 | 0.1315 |

### A.9 MODEL CONFIGURATIONS IN DETAIL

In this section, we describe the network architecture, training setup and evaluation protocol to enable full reproducibility. In addition, we use the FlashAttention-v2 included with Pytorch version 2.2.

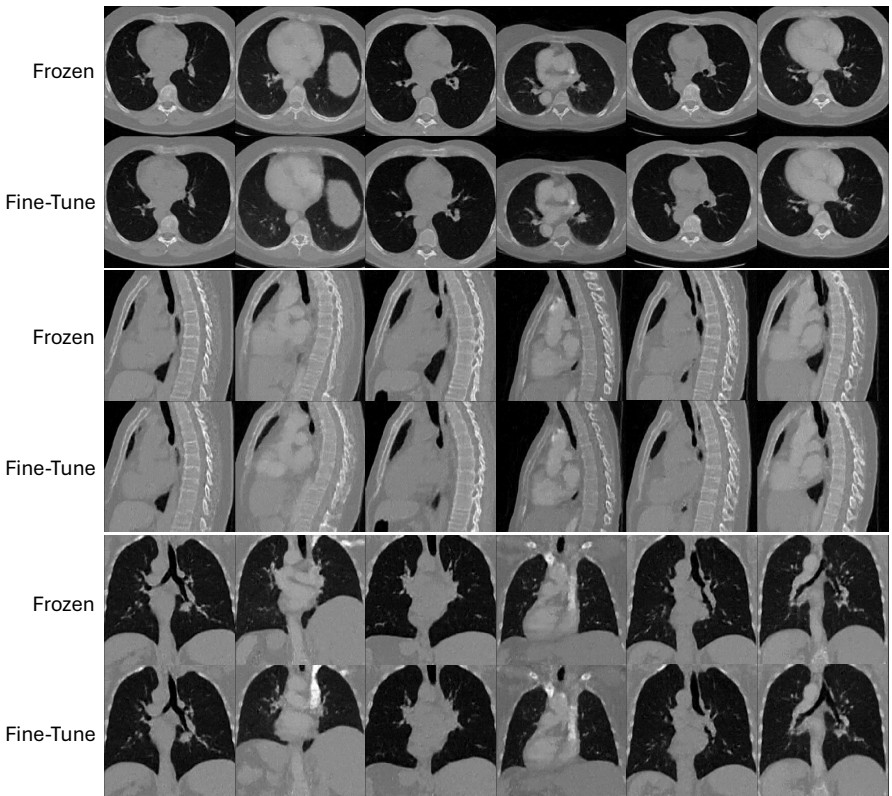

Figure 11: Ablation on freezing vs. fine-tuning the low-resolution backbone at $128^3$ on LIDC-IDRI. Fine-tuning with a small learning rate yields slightly better FID/MMD, while the frozen backbone already achieves competitive performance.

### A.9.1 NETWORK ARCHITECTURE CONFIGURATIONS

The PRDiT model comprises two primary components: a Local Denoiser (MLP-based) and an Global Residual DiT models (Transformer-based), enabling a two-stage training process. Stage 1 trains the coarse path (depth=0), while stage 2 freezes the local path and trains the global path (depth>0). The architecture processes 3D volumes of shape $[B, C, D, H, W]$ into patch sequences for denoising. Key components are described below, with parameters summarized in Table 9, 10.

| Parameter | Value |
| --- | --- |
| Input Channels ($C_{\text{in}}$) | 1 |
| Patch Size ($P \times P \times P$) | $12 \times 12 \times 12$ |
| Stride | 8 |
| Padding | 2 |
| MLP Structure | Two-layer SwiGLU |
| Hidden Size | $C_{\text{in}} \cdot P^3 = 1 \cdot 12^3 = 1728$ |
| MLP Ratio ($R_{\text{mlp}}$) | 1.0 |
| Output Channels ($C_{\text{out}}$) | 2 |
| Normalization | LayerNorm |
| Conditioning | AdaLN_modulation with TimeEmbedding |
| Activation | SwiGLU |
| Dropout | 0.0 |
| Weight Initialization | Zero-initialized for AdaLN and final linear layer |

Table 9: Configurations of the `LocalDenoiser` module for PRDiT.

| Parameter | Value |
|---|---|
| Input Channels ($C_{\text{in}}$) | 1 |
| Patch Size ($P \times P \times P$) | $12 \times 12 \times 12$ |
| Stride | 8 |
| Padding | 2 |
| Depth | $\{4, 8, 12\}$ |
| Hidden Size ($D_{\text{hidden}}$) | $\{768, 1152\}$ |
| Number of Heads | $\{12, 16\}$ |
| MLP Ratio ($R_{\text{mlp}}$) | 4.0 |
| Output Channels ($C_{\text{out}}$) | 2 |
| Normalization | LayerNorm |
| Positional Embedding | 3D sinusoidal, shape $[1, N, D_{\text{hidden}}]$, non-learnable |
| Conditioning | AdaLN-Modulation with TimeEmbedding |
| Attention | Multi-head self-attention with memory-efficient attention |
| Dropout | 0.0 |
| Weight Initialization | Zero-initialized for AdaLN-Zero and final linear layer |

Table 10: Configurations of the `GlobalResidual DiT` module (global residual path) for PRDiT, with depths 4, 8, and 12. All models share the same pretrained `LocalDenoiser` model from Stage 1 training.

### A.9.2 TRAINING SETUP

We provide the details of how to train the methods.

- **Optimization Function**: AdamW with $\beta_1 = 0.9$, $\beta_2 = 0.999$, and weight decay 0.0. The objective is mean squared error (MSE) loss.
- **Learning Rate**:
  - Stage 1 (`LocalDenoiser`): $10^{-4}$.
  - Stage 2 (`GlobalResidual DiT`):
    * $9 \times 10^{-5}$ (depth=4),
    * $6 \times 10^{-5}$ (depth=8),
    * $4 \times 10^{-5}$ (depth=12)

    for new parameters; pretrained parameters (Local Denoiser) are frozen during second-stage training.
- **Device**: NVIDIA A100 80GB GPUs with BFP16 mixed precision.
- **Batch Size**: 128 volumes (16 per GPU), 256 volumes (4 per GPU).
- **Diffusion Schedule**: Linear, 1000 timesteps.
- **Training Steps**: Stage 1: 6,000 epochs; Stage 2: 8,000 epochs.
- **Gradient Clipping**: Maximum norm of 1.0.
- **Data Split**: Randomly-sampled data split of 90% for actual training and a 10% validation set for tracking progress (both datasets).

### A.9.3 EVALUATION PROTOCOL WITH FOCUS ON THE W-SCORE METRIC

In here, we provide the details about how to measure the performance of the generative models. In this paper, we use the three evaluation metrics, 3D FID (Friedrich et al., 2024), 3D MMD (Sun et al., 2022), and W-GAN scores (Zhang et al., 2025). We explain how to use the WGAN-GP Critic scores to measure the generative quality in our experiments.

**The Goal.** We use a single 1-Lipschitz critic $f_\theta$ (WGAN-GP) as a measurement to estimate the Wasserstein-1 distance between the real data distribution $\mathbb{P}_{\text{real}}$ and a model $m$'s distribution $\mathbb{P}_m$:

$$\widehat{W}_1(\mathbb{P}_{\text{real}}, \mathbb{P}_m) = \mathbb{E}_{x \sim \mathbb{P}_{\text{real}}}\big[f_\theta(x)\big] - \mathbb{E}_{\tilde{x} \sim \mathbb{P}_m}\big[f_\theta(\tilde{x})\big]. \tag{9}$$

Lower is better. We compare models $m \in \{$HA-GAN, 3D-LDM, WDM-3D, PRDiT-4L (ours), PRDiT-8L (ours)$\}$ against our PRDiT-12L as the best performance standard.

**Data Splits & Preprocessing.**

1. **Real**: Split the dataset into *Real-Train* and *Real-Val* (no overlap with any generator's training data).

2. **Fake per model**: For each generator $m$, split its generated volumes into *Fake-Train*$^{(m)}$ and *Fake-Val*$^{(m)}$ (no overlap).

3. **Preprocessing**: Apply identical resampling/cropping and intensity normalization (e.g., HU $\rightarrow [0, 1]$) to both real and fake data, during training, it will map $[0, 1] \rightarrow [-1, 1]$.

4. **Mixed Fake for training**: Build *Fake-Train-Mix* by combining the pairs of *Fake-Train*$^{(m)}$ sets.

**Critic & Training Objective.** A 3D conv stack with stride-2 downsamples, InstanceNorm3d and LeakyReLU activations, which is following the 2D WGAN-GP (Gulrajani et al., 2017) define. *No* sigmoid at the end. In addition, the objective (maximize)

$$\mathcal{L}_{\text{critic}} = \mathbb{E}_{x \sim \text{Real-Train}}[f_\theta(x)] - \mathbb{E}_{\tilde{x} \sim \text{Fake-Train-Mix}}[f_\theta(\tilde{x})] - \lambda \, \text{GP}(\theta), \tag{10}$$

with gradient penalty GP on random interpolates and setting $\lambda = 10$. The default configurations is Adam ($\text{lr} = 10^{-5}$, $\beta_1 = 0.5$, $\beta_2 = 0.999$). The batch size is 40 real / 40 fake, and the fake samples are consisting of the generative model and our anchor PRDiT-12L output.

**Pair-Comparisons vs. PRDiT-12L (Anchor)** Given *PRDiT-12L* is the anchor (chosen by 3D FID/MMD), for each competitor $c$:

$$r_c = \frac{\widehat{W}_1(c)}{\widehat{W}_1(\text{Ours-12L})} \quad (>1 \Rightarrow \text{PRDiT-12L better}). \tag{11}$$

Our experiments report the mean ratio and std over three different random seeds with the trained critic.

### A.10    COMPREHENSIVE PAIRWISE COMPARISONS ACROSS ANCHORS

In this section, we propose to choose the different model as the anchor to compute the WGAN ratio score, and for better visualization, we use the log of WGAN ratio. Using the log ratio makes results signed and interpretable (positive = better than the column model; negative = worse). The results show in Table 11.

### A.11    ADDITIONAL QUALITATIVE INSIGHTS

Here we present additional qualitative examples that complement the quantitative results for the LIDC-IDRI dataset in Figure 12 and Rad-chestCT dataset in Figure 13 for our 4-layer, 8-layer and 12-layer models.

#### A.11.1    LIDC-IDRI

We provide additional generated 3D CT images for our PRDiT models with 4, 8 and 12 layers trained on LIDC-IDRI dataset in Figure 12.

#### A.11.2    RAD-CHESTCT

We provide additional generated 3D CT images for our PRDiT models with 4, 8 and 12 layers trained on Rad-ChestCT dataset in Figure 13.

#### A.11.3    COMPARISONS ON $256^3$ RESOLUTION

We provide the additional generated samples across the HA-GAN, WDM-3D, and our PRDiT-4L with the $256^3$ LIDC-IDRI dataset, in Figure 14.

| | HA-GAN | 3D-LDM | WDM-3D | PRDiT-4L | PRDiT-8L | PRDiT-12L |
|---|---|---|---|---|---|---|
| **HA-GAN** | 0 ($\pm 0$) | -0.302 ($\pm 0.002$) | -1.950 ($\pm 0.055$) | -0.969 ($\pm 0.036$) | -1.199 ($\pm 0.095$) | -1.433 ($\pm 0.051$) |
| **3D-LDM** | 0.302 ($\pm 0.002$) | 0 ($\pm 0$) | -1.602 ($\pm 0.069$) | -0.753 ($\pm 0.043$) | -0.963 ($\pm 0.031$) | -1.045 ($\pm 0.068$) |
| **WDM-3D** | 1.950 ($\pm 0.055$) | 1.602 ($\pm 0.069$) | 0 ($\pm 0$) | -0.142 ($\pm 0.055$) | -0.208 ($\pm 0.080$) | -0.371 ($\pm 0.049$) |
| **PRDiT-4L** | 0.969 ($\pm 0.036$) | 0.753 ($\pm 0.043$) | 0.142 ($\pm 0.055$) | 0 ($\pm 0$) | -0.011 ($\pm 0.003$) | -0.104 ($\pm 0.008$) |
| **PRDiT-8L** | 1.199 ($\pm 0.095$) | 0.963 ($\pm 0.031$) | 0.208 ($\pm 0.080$) | 0.011 ($\pm 0.003$) | 0 ($\pm 0$) | -0.073 ($\pm 0.011$) |
| **PRDiT-12L** | 1.433 ($\pm 0.051$) | 1.045 ($\pm 0.068$) | 0.371 ($\pm 0.049$) | 0.104 ($\pm 0.008$) | 0.073 ($\pm 0.011$) | 0 ($\pm 0$) |

Table 11: Pairwise log-WGAN ratio comparisons. For every model pair $(i, j)$, we report $\mu \pm \sigma$ of the log ratio between the WGAN critic scores for samples from model $i$ and model $j$ (averaged over 4 evaluations). Values $> 0$ indicates the row model is judged more realistic than the column model, $< 0$ indicates the opposite. The matrix is anti-symmetric $(s_{i,j} = -s_{j,i})$ with zeros on the diagonal.

### A.11.4 EXTENDED QUALITATIVE RESULTS

We present qualitative results comprising the Local Denoiser branch's coarse reconstructions and the Global Residual DiT branch's residual estimates, highlighting the complementary information each branch captures, in Figure 15.

(a) **4-Layer Model Generative Output**

(b) **8-Layer Model Generative Output**

(c) **12-Layer Model Generative Output**

Figure 12: Extended qualitative results for different model within LIDC-IDRI dataset.

(a) **4-Layer Model Generative Output**

(b) **8-Layer Model Generative Output**

(c) **12-Layer Model Generative Output**

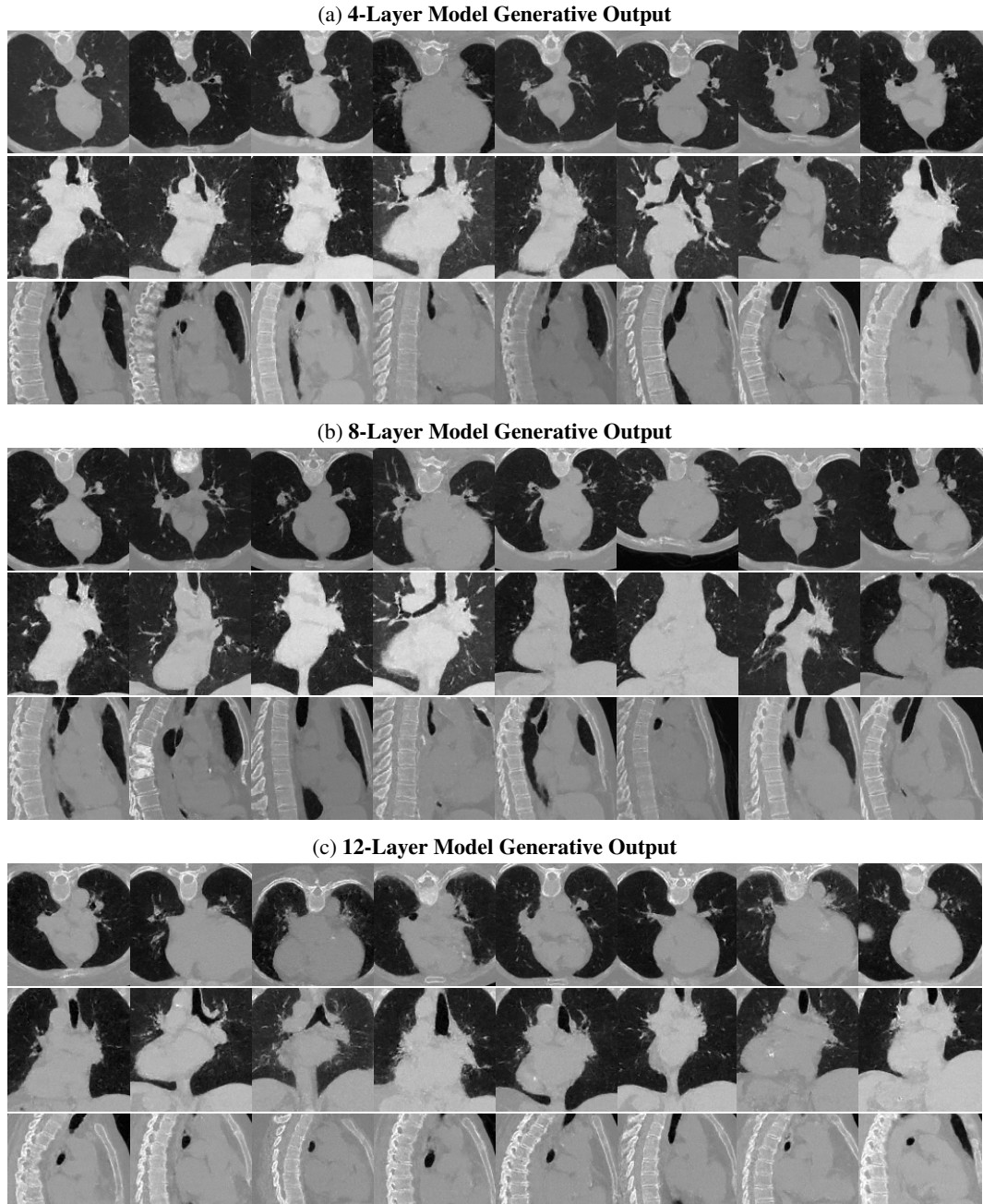

Figure 13: Extended qualitative results for different model within Rad-ChestCT dataset.

(a) **HA-GAN Model Generative Output**

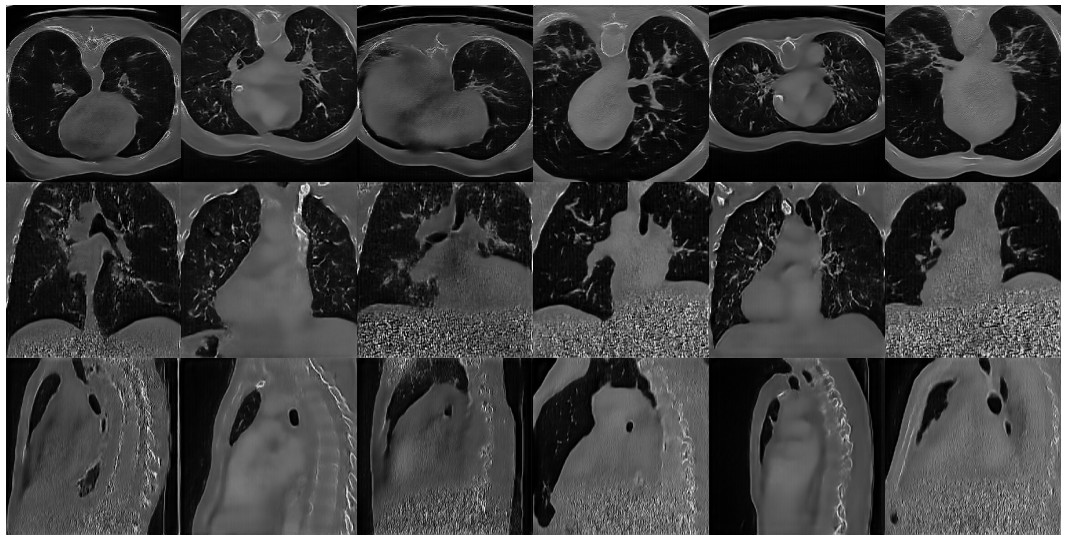

(b) **WDM-3D Model Generative Output**

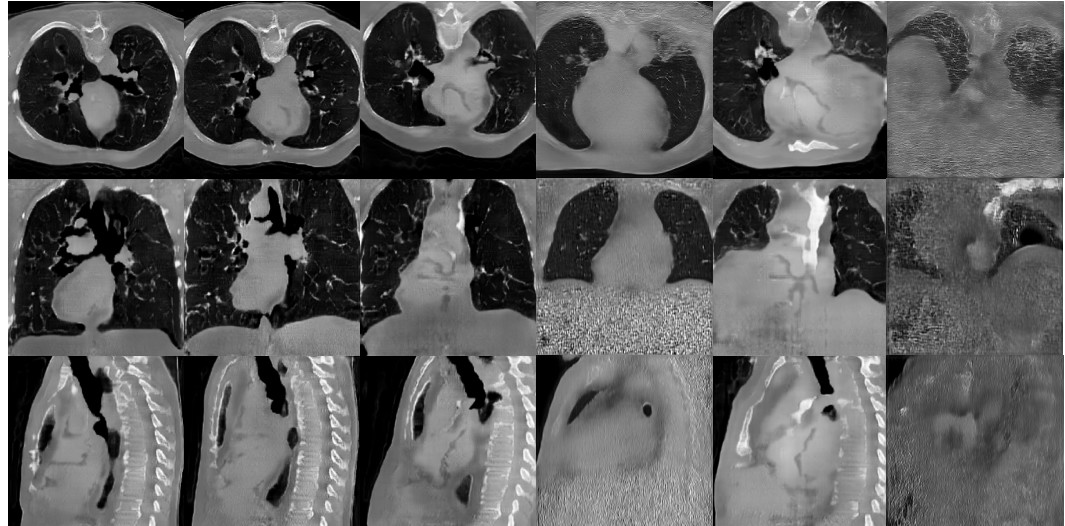

(c) **Our Model Generative Output**

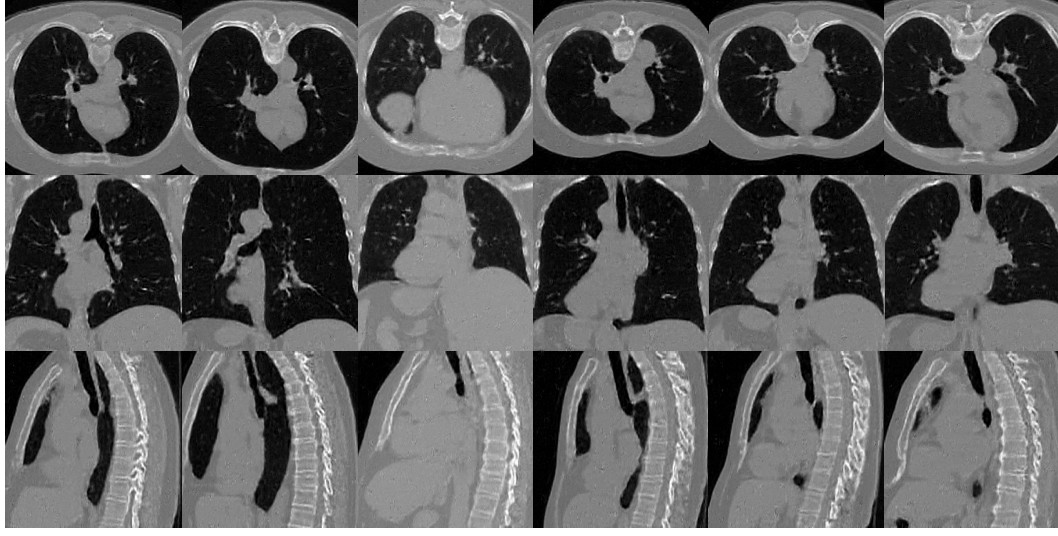

Figure 14: Extended qualitative results for different model within $256^3$ LIDC-IDRI dataset.

(a) **Local Denoiser Path Output**

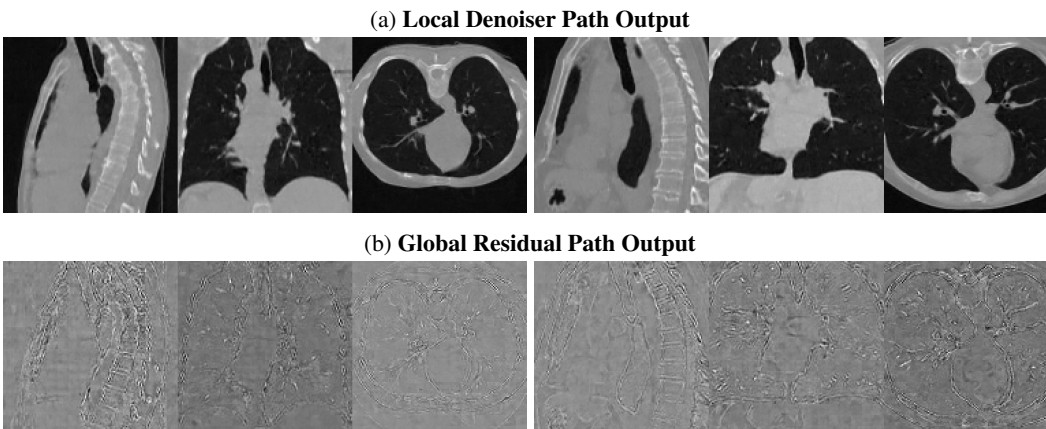

(b) **Global Residual Path Output**

Figure 15: Extended qualitative results for visualizing the output from Local Denoiser and Global Residual module.

