# OpenReview forum: "Pixel-Level Residual Diffusion Transformer: Scalable 3D CT Volume Generation"
_ICLR.cc/2026/Conference — ICLR 2026 Poster_

### Official Review · Reviewer_Vjha · 2025-10-16

**Soundness:** 3
**Presentation:** 3
**Contribution:** 3
**Rating:** 4
**Confidence:** 4

**Summary:**

This paper proposes a transformer-based 3D diffusion model for medical CT generation that integrates a patch-wise Local Denoiser and a global Diffusion Transformer within a two-stage residual framework. A predictor–corrector (“hot diffusion”) sampling strategy improves generation stability and diversity. Comprehensive ablation studies validate each component and analyze computational efficiency. The progressive training scheme reuses a pretrained low-resolution model and fine-tunes lightweight residual modules at higher resolutions, achieving competitive performance with reduced training cost.

**Strengths:**

1. The progressive training strategy from low to high resolution is clearly defined and achieves comparable performance with reduced computational cost.

2. The combination of local and global components provides a balanced design for modeling fine details and overall structure.

3. The model attains competitive quantitative results and produces visually consistent high-resolution 3D CT volumes.

**Weaknesses:**

1. Visual inconsistency in generated samples:
In Figure 5, the second and fifth examples under Ours show black voids in the lower-left region, indicating low-density areas that are absent in the real data. The authors should clarify whether these artifacts result from the “hot diffusion” sampling strategy, which may disrupt local density continuity or anatomical consistency.

2. Extra dataset evaluation:
Additional experiments on CT datasets such as CT-ORG are recommended to further validate the model’s generalization.

3. Potential limitation of frozen local denoiser:
The local denoiser remains frozen during high-resolution training. It would be valuable to include an ablation comparing frozen versus fine-tuned local modules to assess potential performance degradation at higher resolutions.

**Questions:**

1. What is the potential to extend this framework to conditional generation, such as generating raw CT volumes from segmentation maps or other structural priors?

2. In Figure 5, several artifacts are visible. Could the authors explain their causes and further evaluate the influence of the parameter k on generation quality through additional qualitative or quantitative analysis?

3. The local denoiser is frozen during high-resolution training. Could this design lead to performance degradation at higher resolutions? An ablation comparing frozen and fine-tuned local denoisers would help clarify this effect.

---

> ### Author Response · Authors · 2025-11-20
> **Response to Reviewer Vjha (part 1/2)**
>
> We thank you for your helpful suggestions and in-detail inspection of the visuals, and address your points in the following.
>
> **[W1] Visuals**
> > *Visual inconsistency in generated samples [...] low-density areas that are absent in the real data [...]*
>
> It is important to first note that the images in Fig. 5 cannot be compared in a 'per-column' manner, as they are randomly generated samples and not conditioned on any particular ground-truth. (Also see response to Reviewer *Neuf* [W1].)
>
> $\rightarrow$ The dark 'voids' highlighted in the 2nd and 5th examples of Fig. 5 actually correspond to *low-density air cavities* that are also *present in the real data*, rather than artifacts introduced by the hot-diffusion sampler or hallucination issues.
> $\rightarrow$ Following standard practice in prior works, our pre-processing clamps HU values to a fixed window, which further accentuates such air regions as dark areas in both real and generated scans.
>
> $\Rightarrow$ To improve clarity around this, we have added a range of representative slices from the LIDC-IDRI (left, including case-IDs) and Rad-ChestCT (right) training datasets as Fig. 6 to Appendix A.4 of our revised manuscript. We use red boxes to highlight real cases where similar black, low-density regions appear in the lower part of the thorax/upper abdomen (similar to the ones present in the images you pointed out). Their shape, location and surrounding anatomy closely match the structures seen in our generated samples in Figure 5.
>
> $\Rightarrow$ Thus, the 'voids' are indeed consistent with plausible anatomical patterns in the training distribution and are not discontinuous structures that would indicate instability of the hot-diffusion sampling strategy.
>
> ---
> **[W2] Additional dataset**
> > *Additional experiments on CT-ORG*
>
> Although we conducted some preliminary experiments on the data, we ultimately found it unsuitable as a benchmark for our unconditional high-resolution 3D generative setting for the following reasons:
>
> $\rightarrow$ CT-ORG is quite *small* for training and evaluating deep models: it contains only 140 CT volumes in total, with 119 scans for training and 21 scans for evaluation, whereas LIDC-IDRI and Rad-ChestCT splits contain 1,010 and 3,630 scans, respectively.
> $\rightarrow$ In addition, the scan regions *vary substantially* across subjects (full-body or partial torso), which introduces large variation in anatomy and field-of-view, and further reduces the already low 'coverage' of particular parts of the body.
> $\rightarrow$  With only 21 test samples, distributional metrics such as FID, MMD or W-distance become extremely noisy and effectively non-informative. We therefore believe that any numbers reported on this dataset would not be statistically meaningful.
>
> $\Rightarrow$ This dataset seems primarily designed for segmentation with voxel-wise labels, and is indeed very valuable for this purpose. However, its size and heterogeneity make it unsuitable as a basis for training and benchmarking the unconditional 3D generative modeling task that we consider in this work.
>
> ---
> **[W3] Frozen vs. fine-tuned backbone**
> > *ablation comparing frozen versus fine-tuned*
>
> We agree that an explicit ablation between a frozen and a fine-tuned lower-resolution backbone is a very informative addition.
> $\rightarrow$ We conducted experiments at $128^3 \to 256^3$, directly comparing our model with a frozen low-resolution backbone to one where we fine-tune the entire backbone, and evaluate it under exactly the same setup as the frozen-backbone model.
>
> As the table below shows, the fine-tuned backbone yields modest but consistent gains over the frozen one (better FID and MMD), indicating that full fine-tuning (including the backbone) can indeed help to further improve results:
> | Low-res backbone | FID ↓ | MMD ↓ |
> |--|--|--|
> | Frozen $128^3$ | 2.280  | 0.1370  |
> | Fine-tuned $128^3$ | 1.702  | 0.1315  |
>
> $\Rightarrow$ These improvements, however, come with the extra computational cost required for full fine-tuning (+3GPUh for our experiment), and it is ultimately up to the end-user to weigh whether the additional compute is justified for their use case. This tradeoff also motivated our choice to keep the backbone frozen in the main $256^3$ experiments.
>
> $\Rightarrow$ We additionally provide qualitative insights regarding the differences in generative image quality in Appendix A.8 (Fig. 11).
>
> We have added this ablation, along with the corresponding details and hyperparameter choices, to the revised manuscript in Appendix A.8.1. (We are also considering integrating this ablation into the main paper should space permit, as it indeed provides useful additional insight.)

---

> > ### Author Response · Authors · 2025-11-20
> > **Response to Reviewer Vjha (part 2/2)**
> >
> > **[Q1] Extension to conditional generation**
> > > *What is the potential to extend this framework to conditional generation, such as generating raw CT volumes from segmentation maps or other structural priors?*
> >
> > Extending our model/setup to conditional generation appears technically feasible, and we agree that this is a very interesting direction.
> > $\rightarrow$ However, as we indicated in our 'Future Work' section in the Appendix, a thorough investigation of different conditional setups is outside the scope of the present work, which focuses on the challenging unconditional high-resolution and dense 3D medical setting.
> >
> > $\rightarrow$ We hope that future work (either by us or the broader community) can build on our framework to develop and study such conditional variants, and we intend to make our code publicly available to facilitate further research.
> >
> > ---
> > **[Q2a] Visuals**
> > > *In Figure 5, several artifacts are visible. Could the authors explain their causes [...]*
> >
> > Please see our response to [W1].
> >
> > ---
> > **[Q2b] Influence of $k$**
> > > *Could the authors [...] further evaluate the influence of the parameter k on generation quality through additional qualitative or quantitative analysis?*
> >
> > Table 4 (right) in the paper provides a quantitative ablation on varying $k$: as $k$ increases from $1$ (cold sampling) to $3$, both FID and MMD steadily improve, while further increasing $k$ to $4$ starts to hurt performance.
> > $\rightarrow$ Intuitively, $k$ controls how much stochastic exploration is injected by the predictor-corrector steps: small $k$ behaves almost deterministically and tends to under-explore the data manifold, whereas very large $k$ adds excessive noise and degrades image quality. Moderate values such as $k=2$-$3$ strike a good balance between denoising and exploration.
> >
> > $\Rightarrow$ To complement these quantitative insights, we have now added qualitative results to Appendix A.6 (Figs. 7, 8, 9), showing axial, coronal, and sagittal views generated for different $k$.
> >
> > $\Rightarrow$ Consistent with the quantitative trends, $k=1$ samples appear either slightly over-smooth or noisy, $k=2$-$3$ produce sharper structures and clearer vessels, whereas $k=4$ starts to introduce visible artifacts.
> >
> > ---
> > **[Q3] Frozen vs. fine-tuned backbone**
> > > *ablation comparing frozen and fine-tuned*
> >
> > Please see our response to **[W3]**.
> >
> > ---
> > ---
> > We hope our answers have clarified your questions.
> > If any further queries remain, please let us know and we will do our best to promptly address them.

---

> > > ### Comment · Reviewer_Vjha · 2025-11-23
> > >
> > > The authors have provided clear clarifications and additional ablation experiment. The dark regions in the generated samples are explained as low-density air cavities commonly present in real CT scans, supported by examples added to the appendix. The decision not to include CT-ORG results is reasonably motivated by the dataset’s limited size and heterogeneous coverage. The newly added comparison between frozen and fine-tuned backbones is helpful and shows modest improvements together with a clear discussion of computational cost. Overall, the provided responses adequately resolve the specific questions I raised.

---

> ### Author Response · Authors · 2025-11-25
> **Thanks for your support!**
>
> We are glad to hear that our responses have resolved your questions!
>
> Thank you again for your helpful feedback, and for increasing your score.

---

### Official Review · Reviewer_4BSb · 2025-10-30

**Soundness:** 3
**Presentation:** 3
**Contribution:** 3
**Rating:** 6
**Confidence:** 4

**Summary:**

The paper introduces PRDiT, a two-stage diffusion transformer framework for synthesizing high-resolution 3D CT volumes directly at voxel level. A predictor–corrector diffusion sampling method and a progressive low-to-high-resolution training strategy improve sample fidelity and efficiency. Experiments on LIDC-IDRI and Rad-ChestCT show clear advantage over other methods.

**Strengths:**

1. The decomposition of diffusion into local + global residual branches is elegant and addresses the long-standing trade-off between local detail and global coherence in 3D image synthesis;
2. The low-to-high-resolution reuse strategy reduces training cost
3.Figures 4–5 demonstrate noticeably sharper bone edges, smoother organ boundaries, and fewer artifacts relative to baselines
4. Strong reproducibility section (datasets, configs, metrics) and detailed appendices on architecture, hyperparameters, and inference time

**Weaknesses:**

The idea of splitting local/global branches is incremental relative to prior hierarchical or multi-scale diffusion models. The architectural originality lies mainly in combining them via residual refinement rather than introducing new attention or tokenization mechanisms.

Reported mean ± std over 3 seeds is small; given large variance in 3D generation, stronger statistical support or significance tests would enhance credibility.

No mention of hallucination risk or downstream misuse

**Questions:**

n/a

---

> ### Author Response · Authors · 2025-11-20
> **Response to Reviewer 4BSb**
>
> We thank you for your constructive feedback and address your queries individually in the following.
>
> **[W1] Architectural novelty**
> > *[...] incremental relative to prior hierarchical [...]. The architectural originality lies mainly in combining them via residual refinement*
>
> While we are currently not aware of any works that have employed such an hierarchical/hybrid approach in the context of dense medical 3D generation, we fully acknowledge that our model is built using concepts from the diffusion model and efficiency literature (correspondingly cited in the paper).
> $\rightarrow$ The goal of our work is not to introduce a completely new primitive; The crucial point is that *neither the local MLP branch nor the global DiT alone* can handle this regime effectively: the former lacks global context, and the latter would be prohibitively expensive at full 3D resolutions.
> $\rightarrow$ We demonstrate that our hybrid architecture, where the MLP performs the 'heavy' local denoising and the DiT supplies only global residual refinements, is sufficient to achieve globally coherent and anatomically detailed CT volumes while staying within realistic memory limits.
> $\rightarrow$ To the best of our knowledge, this specific decomposition has not been explored for dense 3D medical data, and our experiments and results illustrate that it meaningfully expands what is practically trainable in this domain.
>
> ---
> **[W2] Statistics/Robustness**
> > *mean ± std over 3 seeds is small*
>
> To further validate robustness, we have now run our main LIDC-IDRI experiments with two additional random seeds and recomputed the statistics over $5$ seeds.
> | Model | FID ↓ | MMD ↓ |
> |--|--|--|
> | HA-GAN | 3.07 ± 0.34 | 0.1976 ± 0.008 |
> | 3D-LDM | 7.46 ± 0.45 | 0.3607 ± 0.015 |
> | WDM-3D | 3.55 ± 0.36 | 0.1881 ± 0.016 |
> | PRDiT-4L (ours) | 2.01 ± 0.14 | 0.1825 ± 0.010 |
> | PRDiT-8L (ours) | 1.90 ± 0.10 | 0.1648 ± 0.010 |
> | **PRDiT-12L (ours)** | **1.48 ± 0.16** | **0.1461 ± 0.008** |
>
> $\Rightarrow$ Results change only slightly when compared to the original 3-seed ones in Table 1 of our paper; e.g. PRDiT-12L moves from $1.41 \pm 0.17$ to $1.48 \pm 0.16$ FID and from $0.1501 \pm 0.010$ to $0.1461 \pm 0.008$ MMD, while still clearly outperforming the baselines.
> This re-confirms that random-seed variance is small relative to the performance margins.
>
> $\Rightarrow$ We are currently running the additional seeds also for the other dataset, and will update the Tables in the revised manuscript with the 5-seed results once the experiments are completed.
>
> ---
> **[W3] Hallucinations and Risks**
> > *No mention of hallucination risk or downstream misuse*
>
> In our context, 'hallucination’ would refer to anatomically implausible or extremely rare structures that lie far outside the training distribution.
>
> From a technical standpoint, we monitor this risk partly via the W-score metric:
> $\rightarrow$ If the model frequently produced unrealistic volumes, the WGAN critic would assign larger distances between real and generated samples, and the W-score would deteriorate.
> $\rightarrow$ In practice, our W-scores remain close to those of real data, which suggests that severe hallucinations are uncommon in our experiments.
>
> $\Rightarrow$ That said, we do *not* position PRDiT as a tool for direct clinical decision-making. Synthetic CT volumes should be used only for research, simulation, or carefully controlled data augmentation, and always under expert supervision. Any downstream clinical application must rely on appropriate validation on real patient data and the involvement of qualified radiologists.
>
> We have added a separate 'Potential risks and societal impacts' section (Appendix A.2) in the revised version of our manuscript to make these limitations and the associated societal/ethical considerations more explicit.
>
> ---
> ---
> We hope our answers have addressed your concerns.
> If you have any further questions, please let us know.

---

### Official Review · Reviewer_eQz3 · 2025-11-01

**Soundness:** 3
**Presentation:** 4
**Contribution:** 2
**Rating:** 6
**Confidence:** 5

**Summary:**

I reviewed the paper, a two-stage diffusion architecture for high-resolution 3D medical image synthesis. The model combines a local MLP denoiser, which captures fine-grained voxel details on overlapping patches, with a global residual Transformer, which ensures anatomical consistency across the whole volume.
The authors also propose a predictorcorrector sampling scheme that reintroduces controlled noise to stabilize generation, and a low-to-high-resolution scaling strategy that allows efficient 256 training guided by a pretrained 128 backbone.

They evaluate their method on LIDC-IDRI and Rad-ChestCT, showing improved FID and MMD over several baselines (HA-GAN, 3D-LDM, WDM-3D). The generated CT volumes appear sharper and more realistic, while training remains computationally feasible. Overall, the work presents a solid engineering improvement that makes transformer-based diffusion models more practical for 3D medical data synthesis.

**Strengths:**

I find the paper technically solid. The proposed two-stage design separating local voxel-level denoising from global residual refinement is clearly explained, and supported by ablation studies showing that each component contributes to performance. The predictorcorrector sampling strategy is effective, improving image quality with minimal computational overhead. I also appreciate the scaling approach that leverages a pretrained low-resolution model to enable efficient 256 synthesis, which addresses an important computational bottleneck for 3D diffusion models.

The paper includes comprehensive experiments on two public CT datasets, reports multiple quantitative metrics (FID, MMD, and provides consistent improvements over established baselines. Figures illustrate sharper structural detail and realistic textures. I also value the inclusion of limitations and future work, as well as the clear structure and readability of the paper.

**Weaknesses:**

While the method is good, I find the novelty somewhat limited. The combination of a local denoiser and a global Transformer is conceptually straightforward and resembles existing hierarchical or hybrid DiT approaches. Similarly, the proposed sampling resembles previously known stochastic sampling methods, though applied here in a slightly modified form.

The evaluation focuses mainly on generative metrics such as FID and MMD, which can  be unstable for 3D medical data. There is no downstream or clinical validation (e.g., segmentation or detection performance using synthetic data), which makes it difficult to judge real-world utility. Baselines are also somewhat limited and some stronger 3D diffusion models or DiT variants are missing, especially at higher resolutions

Finally, several implementation details are underspecified, such as the exact memory-efficient attention mechanism, data split protocol, and reproducibility of high-resolution experiments. Overall, the paper’s contribution feels more like a solid engineering refinement than a major conceptual breakthrough.

**Questions:**

- Could you clarify which memory-efficient attention variant is used in the global DiT (e.g., FlashAttention, windowed, or block-sparse)? How much does it contribute to scalability compared to a vanilla DiT?
- Predictor sampling: Can you provide pseudocode or additional explanation of the schedule? Did you explore adaptive or variable numbers of corrective steps (k > 2), and how stable is training when increasing k?
- Baselines: Why were recent efficient 3D diffusion or DiT variants  excluded from comparison? Would your method still outperform these stronger models, especially at 256 resolution?

---

> ### Author Response · Authors · 2025-11-20
> **Response to Reviewer eQz3 (part 1/2)**
>
> We thank you for your constructive review, and address your points individually in the following.
>
> **[W1a] Novelty in architecture**
> > *While the method is good, I find the novelty somewhat limited. [...] resembles existing hierarchical or hybrid DiT*
>
> While we are currently not aware of any works that have employed a similar approach in the context of dense medical 3D generation, we acknowledge that our model is built using concepts from the diffusion model and efficiency literature (as cited in the paper).
> $\rightarrow$ Our contribution is not to introduce a completely new primitive, but to make these tools actually work in the regime of dense, high-resolution 3D medical volumes, which is very different from 2D and 3D sparse settings where DiT-like methods have mostly been used:
> Chest CT scans are full voxel grids at $128^3$--$256^3$ with rich fine-grained anatomy, rendering many existing approaches either infeasible (memory constraints) or subpar (too little detail/information captured).
> $\Rightarrow$ These insights are exactly what motivated our hybrid coarse–fine design: The local MLP path carries most of the denoising load at low cost, while the global DiT only predicts residual corrections; this lets us use a much shallower Transformer without sacrificing global coherence, and makes full-volume 3D training feasible even in memory-constrained setups.
>
> It is of course possible that we might have missed some related work, and we would be grateful if you could provide us with a corresponding reference if this is the case (which would also allow us to provide you with a more detailed and specific answer).
>
> ---
> **[W1b] Novelty of sampling method**
> > *sampling resembles previously known stochastic sampling methods*
>
> Our sampler differs from standard DDIM sampling, and can be viewed as a predictor-corrector variant of the ancestral `hot' sampler:
> $\rightarrow$ instead of a single stochastic update from $x_t$ to $x_{t-1}$, we take $k$ deterministic denoising sub-steps followed by $k-1$ stochastic correction steps, which lets us inject a stronger but controllable amount of noise per outer step.
>
> *Side-by-side comparison:*
> The ablation results in the table below demonstrate that with this design, our hot diffusion (e.g. $k=3$) achieves better FID/MMD than both the deterministic case ($k=1$) and the 'conventional' hot-diffusion sampler, indicating that our proposed modified stochastic schedule is genuinely beneficial rather than just a reparameterization of the standard sampler:
> | $k$ | FID ↓ | MMD ↓ |
> |--|--|--|
> | 1 ('cold') | 7.031| 0.7285 |
> | 2 | 6.132 | 0.6023 |
> | 2.5 | 5.950 | 0.5893 |
> | 3 | 5.736 | 0.5699|
> | 3.5 | **5.718** | **0.5682** |
> | 4 | 6.960 | 0.6273 |
> | *conventional* | 6.456 | 0.6100 |
>
> $\Rightarrow$ In Appendix A.6 of the revision, we also provide qualitative insights (generated samples, A.6.2) as well as the underlying equations (A.6.3) and detailed algorithmic description (A.6.1).
>
> ---
> **[W2a] Generative metrics**
> > *FID and MMD [...] can be unstable*
>
> We agree (despite FID/MMD being standard for evaluation).
> $\Rightarrow$ This is precisely why we additionally evaluate our results using the W-score. This metric directly measures how far the generated distribution is from the real empirical CT distribution, and is more sensitive to structural mismatches than FID/MMD.
>
> > *no downstream or clinical validation*
>
> There exists unfortunately no voxel-wise `ground truth' for novel unconditionally-generated volumes.
> $\rightarrow$ Downstream segmentation or detection studies would require dense, expert annotations for both real and synthetic CT volumes, which is *expensive* and therefore unfortunately beyond the scope and resources of this work.
> $\rightarrow$ The objective of this paper is to make high-fidelity unconditional 3D CT generation feasible at scale and to rigorously evaluate distributional fidelity. We therefore have to leave a systematic study of using our model for downstream segmentation/detection and assessing its clinical impact as future work.
>
> $\Rightarrow$ We have expanded our limitations paragraph in Appendix A.3 to explicitly state that (i) we do not claim immediate clinical deployment, and (ii) such downstream and clinical evaluations are outside the scope of the present paper.

---

> ### Author Response · Authors · 2025-11-20
> **Response to Reviewer eQz3 (part 2/2)**
>
> **[W2b] Baselines**
> > *somewhat limited and some stronger 3D diffusion models or DiT variants are missing*
>
> Our baseline choices are constrained by what is practically feasible for dense 3D CT generation at high resolution and what is publicly available.
> $\rightarrow$ Most existing `3D DiT'-like works either operate only on 2D slices or on surfaces / triplane representations.
> The recent 3D diffusion method TCAM-Diff (2025, no code available) uses a triplane-aware representation to encode 3D volumes more efficiently: storing three orthogonal feature planes means the representational cost scales as $\mathcal{O}(L^2)$ instead of the $\mathcal{O}(L^3)$ cost of a full voxel grid. This substantially reduces memory and computation, but also means that the model never operates on a fully dense 3D field in the same way as voxel-based generators.
> $\rightarrow$ This difference in scaling makes dense 3D CTs substantially more challenging than surface/triplane-based 3D tasks, and prevents 'naive' application of many methods.
>
> $\Rightarrow$ We therefore focused our efforts on a controlled comparison against widely used dense 3D medical baselines (HA-GAN, 3D-LDM, WDM-3D) under identical data splits and evaluation metrics.
>
> *If you are aware of any related available baseline that we might have missed, we would appreciate if you could provide a reference.*
>
> ---
> **[W3] Implementation details**
> > *memory-efficient attention*
>
> We use FlashAttention-v2 included with PyTorch version 2.2
> > *data split protocol*
>
> We use a randomly-sampled data split of 90\% for actual training and a 10\% validation set for monitoring progress (both datasets).
> > *reproducibility of high-resolution*
>
> We have expanded Appendix A8 to include the details for the high-resolution experiments to ensure reproducibility.
>
> ---
> **[Q1] Attention and Scalability**
> > *how much does [efficient attention] contribute to scalability*
>
> *Scaling through our design:*
> Thanks to our hybrid structure, the global DiT only predicts residual corrections on top of the MLP-based local prior allowing us to keep the Transformer relatively shallow, which reduces memory and compute and improves throughput, and supports scalability.
>
> *Scaling through efficient attention:*
> FlashAttention further lowers the per-layer memory footprint of self-attention, which is critical at $128^3$ - $256^3$ resolutions.
> $\rightarrow$ Vanilla attention would require significant reduction of DiT layers (leading to worse FID/MMD) or shrinking the batch size (slow or infeasible training).
>
> $\Rightarrow$ The scalability of our approach therefore stems from the combination of (i) the residual-only DiT design and (ii) the use of FlashAttention.
>
> ---
> **[Q2] Predictor sampling**
> > *provide pseudocode*
>
> We have added a detailed algorithmic description to Appendix A.6.1 (Algorithm 1) in the revised manuscript, as well as qualitative insights for varying values of step-size $k$ (A.6.2).
> > *explore adaptive or variable numbers of corrective steps*
>
> Table 4 (right) shows the development of MMD and FID for $k\in{1,2,3,4}$ (kept constant during generation), and qualitative insights are presented in the revised Appendix A.6.2.
> We have also briefly experimented with schedules that vary/adapt $k$ throughout the generative process, but did not see major improvements that justified further exploration.
> $\rightarrow$ For insights/comparison with 'conventional' hot sampling and the impact of varying $k$, please also see our response to [W1b].
> > *how stable is training when increasing k*
>
> The step-size $k$ is only used during generation (sampling), not during training.
>
> ---
> **[Q3] Baselines**
> > *recent efficient 3D diffusion or DiT variants*
>
> Please refer to our answer to [W2b].
>
> ---
> ---
> We hope our answers addressed all your questions.
> If you have any further queries, please do not hesitate to reach out.

---

### Official Review · Reviewer_Nuef · 2025-11-08

**Soundness:** 2
**Presentation:** 2
**Contribution:** 2
**Rating:** 6
**Confidence:** 4

**Summary:**

Generating fine-grained 3D CT data is an extremely challenging problem. This paper addresses this challenge by employing a pixel-level residual diffusion transformer. The chosen topic is highly relevant and represents a significant area of research. The methodology utilizes a coarse-to-fine strategy and includes comparisons with various established models.
The authors demonstrated superior performance against existing models by comparing metrics such as 3D DIF, MMD, and W-Score on two generated datasets of $128^3$ resolution. However, a significant drawback is the lack of evaluation by a medical imaging expert from a clinical image generation standpoint.
Observing Figure 4, 5, the model appears to capture the overall coarse shape, but there still seem to be significant problems with the finer details. Specifically, as the layers deepen, internal structures within the lungs appear to vanish, and the soft tissue contrast does not accurately reflect reality.
The overall comparison lacks benchmarking against the current State-of-the-Art (SoTA) DiT (Diffusion Transformer) model. The paper mentions that DiT models suffer from "unstable dynamics and optimization difficulties," yet it fails to provide any comparative experiments to support this claim or justify the exclusion.

**Strengths:**

Generating fine-grained 3D CT data is an extremely challenging problem. This paper addresses this challenge by employing a pixel-level residual diffusion transformer. The chosen topic is highly relevant and represents a significant area of research. The methodology utilizes a coarse-to-fine strategy and includes comparisons with various established models.
The authors demonstrated superior performance against existing models by comparing metrics such as 3D DIF, MMD, and W-Score on two generated datasets of $128^3$ resolution. However, a significant drawback is the lack of evaluation by a medical imaging expert from a clinical image generation standpoint.

**Weaknesses:**

Observing Figure 4, 5, the model appears to capture the overall coarse shape, but there still seem to be significant problems with the finer details. Specifically, as the layers deepen, internal structures within the lungs appear to vanish, and the soft tissue contrast does not accurately reflect reality.

**Questions:**

The overall comparison lacks benchmarking against the current State-of-the-Art (SoTA) DiT (Diffusion Transformer) model. The paper mentions that DiT models suffer from "unstable dynamics and optimization difficulties," yet it fails to provide any comparative experiments to support this claim or justify the exclusion.

---

> ### Author Response · Authors · 2025-11-20
> **Response to Reviewer Nuef**
>
> We thank you for your constructive feedback and the detailed inspection of our figures, and address your points individually in the following.
>
> **[W1] Fine visual detail**
> >  *...problems with the finer details. [...] as the layers deepen, internal structures within the lungs appear to vanish*
>
> The current layout of Figs. 4 and 5 can indeed make it look as if the three columns were `estimates' of the same underlying CT volume.
> $\rightarrow$ This is *not* the case: each column in Fig. 4 is an independent random sample generated by the corresponding model variant (4/8/12 layers), using the same noise seed but not conditioned on any fixed ground-truth CT (and therefore no directly comparable). The columns in Fig. 5 show different samples that are all generated with a 12 layer model.
> $\rightarrow$ The observation that some appear to show less structure is only a property of that specific 'stochastic' sample (and slice), not an indication of a general trend. Please see Appendix Fig. 12 for more qualitative results across depths.
>
> $\Rightarrow$ We will revise the figures' captions to explicitly state that each column corresponds to independent samples which cannot be directly contrasted.
>
> In addition: Note that while such qualitative insights are based on only few samples, the quantitative results in Table 1 provide a more robust view of model behavior over the full 960 evaluation volumes:
> $\rightarrow$ W-score measures the closeness of generated samples to the real data distribution, and loss in detail would result in increased score; yet, increased depth improves W-score, as well as FID and MMD.
>
> ---
>
> **[Q1] Baselines**
> > *[...] comparison lacks benchmarking against the current State-of-the-Art (SoTA) DiT [...]. The paper mentions that DiT models suffer from "unstable dynamics and optimization difficulties," yet it fails to provide [...]*
>
> We acknowledge that our original wording was imprecise:
> $\rightarrow$ DiT variants models have shown very strong performance in *2D image generation*, and there exist some recent 3D DiT-like models for sparse 3D data (shapes or voxelized point clouds) like [Mo et al., NeurIPS 2023].
> $\rightarrow$ However, high-resolution 3D *medical volumes* are a very different regime: dense fields with the millions of voxels and rich fine-scale anatomy.
> $\rightarrow$ Capturing this structure with a standard 3D DiT would require a rather deep Transformer, which further amplifies the inherent memory and optimization burden: memory and compute grows cubically, rendering such naive approaches infeasible for many computational setups.
>
> In our preliminary trials, training such deep 3D DiTs directly on $128^3$–$256^3$ CT volumes was both optimization-fragile and extremely computationally demanding; hence our statement in lines 146/147.
> $\Rightarrow$ These findings were precisely what motivated our hybrid design, where our lightweight local MLP handles most of the coarse denoising and our global 3D DiT only predicts residual corrections, allowing us to use a *much shallower DiT* while keeping high-resolution training stable and affordable.
> $\Rightarrow$ We have modified the corresponding paragraph to emphasize this distinction between sparse (e.g. voxelized point clouds) and dense 3D tasks, and included an additional reference (see lines 145-149).
>
> ---
> ---
>
> We hope our answers have clarified all your questions.
> If you have any further queries, please let us know and we are happy to answer them.

---

> > ### Comment · Reviewer_Nuef · 2025-11-26
> >
> > I have reviewed the revised manuscript and the authors' response letter.
> > The clarification regarding lung is convincing, and the justification for omitting the CT-ORG results is reasonable.
> > Since all my specific queries have been adequately resolved, I recommend the acceptance of this manuscript.

---

> > > ### Author Response · Authors · 2025-11-28
> > > **Thank you for your support**
> > >
> > > We are happy to hear that our answers have resolved your questions, and we thank you for recommending acceptance of our paper.
> > >
> > > If you feel it is appropriate, we would greatly appreciate an updated rating.
> > >
> > > We again thank you for your constructive feedback.

---

### Author Response · Authors · 2025-12-04
**Summary of Rebuttal and Revisions**

We would like to again thank the reviewers for their constructive feedback.
We have revised the manuscript to improve the contextualization of our contributions and included additional experimental evidence to ensure robustness and further improve clarity. We summarize our main responses below:

**Distinct Regime: Dense vs. Sparse 3D (*Nuef, eQz3, 4BSb*):**
Dense medical volumes ($128^3$ - $256^3$ voxel grids) represent a *fundamentally different* computational regime where standard DiTs face a critical trade-off: achieving high fidelity requires sufficient (and often significant) depth, yet the cubic complexity quickly makes deep models computationally intractable.
Our hybrid model design explicitly *redistributes the modeling burden* to make training both feasible and stable:
$\rightarrow$ The local MLP efficiently handles the bulk of the denoising process (in parallel),
$\rightarrow$ which *enables* our global DiT to remain *shallow* (and thus trainable) by focusing strictly on mid-to-long-range residual corrections.
$\Rightarrow$ This decomposition overcomes the depth-feasibility bottleneck, allowing high-fidelity modeling within practical memory limits.

**Methodological Robustness & Ablations (*eQz3, 4BSb, Vjha*):**
- **Robustness/Statistics:** Expanded experiments with 5 random seeds (up from 3) confirm that our performance gains are robust (see revised Table 1, *4BSb*).
- **Superior Sampling:** Quantitative and qualitative ablations demonstrate that our 'hot diffusion' ($k>1$) predictor-corrector sampler outperforms both standard deterministic and conventional 'hot' sampling strategies (see Appendix A.6).
- **Efficiency Validation:** New ablations comparing frozen vs. fine-tuned local backbones in our high-resolution experiments confirm that the frozen-backbone design maintains high fidelity while maximizing training efficiency (see Appendix A.8).

**Anatomical Consistency (*Nuef, Vjha*):**
- **Verified Structures:** The observed 'black voids' (noted by *Vjha*) are in fact anatomical air cavities present in the real training data, not artifacts. Appendix A.4 (Fig. 6) now visually confirms this across both datasets.
- **Independent Samples:** Columns in Figs 4 and 5 represent independent random samples, not incoherent estimates of a single target. Variation across columns reflects diversity, not structural inconsistency (clarified in revised manuscript; also see Appendix Fig. 12).

**Additional Revisions:**
To support reproducibility and further clarify our scope, the revised manuscript now additionally includes:
- A dedicated *Potential Risks and Societal Impacts* section (Appendix A.2), explicitly distinguishing our research methodology from tools intended for direct clinical decision-making;
- Detailed *algorithmic descriptions and ablations* of the sampling schedule (Appendix A.6);
- Extended implementation details and backbone ablations for our high-resolution runs (Appendix A.8); and
- *Real-data evidence* validating the existence of air-cavity structures (Appendix Fig. 6, with annotated sample IDs).

$\Rightarrow$ We hope these clarifications better contextualize the difficulty of the dense 3D regime and reinforce both the novelty and practicality of our efficient 3D medical generative framework.

---

### Meta-Review · Area_Chair_JuQx · 2026-01-06

**Summary:**

This paper presents a novel method for 3D CT synthesis, which deals with a very important problem in medical imaging. Overall, the paper is well written and easy to follow. The reviewers recognized the significance of the work. While some of the concerns on experiments have been raised, the authors have responded with additional results, which have been recognized by two reviewers. Overall, I believe this is a reasonable work for ICLR.

**Reviewer Concerns:**

There are some concerns on the experimental results but the authors have addressed the concerns.

**Reviewer Scores:**

The reviewers have expressed that the rebuttal has addressed some of the concerns. It is possible that the scores would be increased.

---

### Decision · Program_Chairs · 2026-01-26

Accept (Poster)